# Palatal morphology predicts the paleobiology of early salamanders

Jia Jia[1,2,3]*, Guangzhao Li[4], Ke-Qin Gao[1]*

[1]School of Earth and Space Sciences, Peking University, Beijing, China; [2]State Key Laboratory of Palaeobiology and Stratigraphy (Nanjing Institute of Geology and Palaeontology, CAS), Nanjing, China; [3]Department of Comparative Biology and Experimental Medicine, University of Calgary, Calgary, Canada; [4]Department of Microbiology, Immunology, and Tropical Medicine, The George Washington University, Washington D.C., United States

**Abstract** Ecological preferences and life history strategies have enormous impacts on the evolution and phenotypic diversity of salamanders, but the yet established reliable ecological indicators from bony skeletons hinder investigations into the paleobiology of early salamanders. Here, we statistically demonstrate by using time-calibrated cladograms and geometric morphometric analysis on 71 specimens in 36 species, that both the shape of the palate and many non-shape covariates particularly associated with vomerine teeth are ecologically informative in early stem- and basal crown-group salamanders. Disparity patterns within the morphospace of the palate in ecological preferences, life history strategies, and taxonomic affiliations were analyzed in detail, and evolutionary rates and ancestral states of the palate were reconstructed. Our results show that the palate is heavily impacted by convergence constrained by feeding mechanisms and also exhibits clear stepwise evolutionary patterns with alternative phenotypic configurations to cope with similar functional demand. Salamanders are diversified ecologically before the Middle Jurassic and achieved all their present ecological preferences in the Early Cretaceous. Our results reveal that the last common ancestor of all salamanders share with other modern amphibians a unified biphasic ecological preference, and metamorphosis is significant in the expansion of ecomorphospace of the palate in early salamanders.

*For correspondence:
jia.jia@ucalgary.ca (JJ);
kqgao@pku.edu.cn (K-QG)

**Competing interest:** The authors declare that no competing interests exist.

## Editor's evaluation

This paper is a valuable contribution to evolutionary ecomorphology in extant and extinct tetrapods, and of interest to vertebrate paleontologists and other evolutionary biologists interested in the early evolution of amphibians. Using geometric morphometric analysis, the authors demonstrate that both the shape of the palate and several non-shape variables (particularly associated with vomerine teeth) are ecologically informative in early stem- and basal crown-group salamanders. The study also reveals that metamorphosis is significant in the expansion of ecomorphospace of the palate in early salamanders.

## Introduction

Salamanders, anurans, and caecilians are highly distinctive from one another in their morphology in both living species and their respective oldest known relatives from the Triassic (*Ivachnenko, 1978*; *Ascarrunz et al., 2016*; *Pardo et al., 2017a*; *Schoch et al., 2020*; *Kligman et al., 2021*). As a result, the evolutionary origin(s) of modern amphibians have remained controversial since the late 19th century (*Haeckel, 1866*), with a number of extinct tetrapod groups in different ecological types at

adult stages having been hypothesized as their purported ancestors, including: amphibamid (terrestrial) and branchiosaurid (terrestrial and aquatic) dissorophoid temnospondyls (*Laurin et al., 2004*; *Fröbisch and Schoch, 2009*; *Maddin and Anderson, 2012*; *Pardo et al., 2017b*), stereospondylian (semiaquatic/aquatic) temnospondyls (*Schoch and Milner, 2014*; *Pardo et al., 2017a*), and several groups of lepospondyls (aquatic, semiaquatic or terrestrial; *Marjanović and Laurin, 2013*; *Marjanović and Laurin, 2019*; *Jansen and Marjanović, 2021*; *Laurin et al., 2022*). The specialized morphologies in modern amphibians are greatly impacted by ecology and their complex life history strategies (e.g. *Wake, 2009*), for example, even the earliest anuran *Triadobatrachus* and the possible caecilian *Chinlestegophis* from the Triassic display several morphological specializations as their living relatives for aboveground and subterranean terrestrial living settings, respectively. Salamanders (or Caudata, the total group), on the other hand, have a more conservative body plan and more diversified ecological preferences when compared to anurans and caecilians (*Deban and Wake, 2000*; *Bonett and Blair, 2017*; *Fabre et al., 2020*), and have been frequently used as comparative analogues for inferring the paleoecology of extinct tetrapods (*Schoch and Fröbisch, 2006*; *Fröbisch and Schoch, 2009*). However, the evolutionary paleoecology in modern amphibians and particularly in early salamanders has received insufficient attention.

Cryptobranchoidea is the sister group of all other crown group salamanders (Urodela) and contains two subclades: Pancryptobrancha (total group cryptobranchids; *Vasilyan et al., 2013*) and Panhynobia (total group hynobiids; *Jia et al., 2021a*). The two subclades are united by a set of synapomorphies (*Dunn, 1922*; *Estes, 1981*; *Jia et al., 2021a*), but are different from each other in life history strategies and ecological preferences at their respective adult stage: most pancryptobranchans are neotenic or partially metamorphosed and live in water permanently by retaining larval features (e.g. gill slits), albeit the pancryptobranchan *Aviturus* from the Paleocene was interpreted as semiaquatic with an unknown life history strategy (*Vasilyan and Böhme, 2012*; but see *Skutschas et al., 2018*). In contrast, panhynobians are predominantly metamorphosed, except that the stem hynobiid *Regalerpeton* from Early Cretaceous (*Rong, 2018*) and some populations of the living hynobiid *Batrachuperus londongensis* are neotenic (*Jiang et al., 2018*). Postmetamorphosed hynobiids have lost larval features and are characterized by an anterolaterally directed palatal ramus of the pterygoid, and are able to live in water (e.g. *Paradactylodon*), on land (e.g. *Hynobius*) or are semiaquatic (*Ranodon*) outside of the breeding season (*Kuzmin and Thiesmeier, 2001*; *Fei et al., 2006*; Materials and methods).

Cryptobranchoidea are critical in understanding the paleoecology of early salamanders because the earliest known cryptobranchoids from the Middle Jurassic (Bathonian) have higher disparities in both life history strategies and ecological preferences than stem urodeles, and represent the oldest known crown urodeles, including 'Kirtlington salamander B' from the UK, *Kiyatriton krasnolutskii* from Russia, and *Chunerpeton*, *Neimengtriton*, and *Jeholotriton* from China (*Evans and Milner, 1994*; *Gao et al., 2013*; *Skutschas, 2015*; *Jia et al., 2021a*). Both *Chunerpeton* and *Jeholotriton* are neotenic as confirmed by the presence of external gills and a tall caudal dorsal fin in adult specimens (*Gao and Shubin, 2003*; *Wang and Rose, 2005*), whereas *Neimengtriton* is the oldest metamorphosed and semiaquatic cryptobranchoid (*Jia et al., 2021a*; see below). The 'Kirtlington salamander B' and *K. krasnolutskii* are both represented by fragmentary materials and their paleoecology unfortunately remains unknown. In contrast, other contemporaries (e.g. *Kokartus*, *Marmorerpeton*) from the Middle Jurassic (Bathonian) of UK, Russia, and Kyrgyzstan are all neotenic and aquatic at their adult stage, and have been classified as stem urodeles by the absence of spinal nerve foramina in the atlas that characterizes Urodela (*Ivachnenko, 1978*; *Evans et al., 1988*; *Skutschas and Krasnolutskii, 2011*; *Skutschas and Martin, 2011*; *Skutschas, 2013*; *Skutschas et al., 2020*). The only known pre-Jurassic stem urodele, *Triassurus* from the Middle/Upper Triassic of Kyrgyzstan, is merely represented by two larval specimens with no clue to its paleoecology at adult stage (*Schoch et al., 2020*).

To date, seven other basal cryptobranchoids have been reported from the Upper Jurassic to Lower Cretaceous of northern China: *Laccotriton*, *Liaoxitriton*, *Linglongtriton*, *Nuominerpeton*, *Pangerpeton*, *Regalerpeton,* and *Sinerpeton*, most of which are represented by articulated specimens and have been recently recovered as stem hynobiids or hynobiid-like taxa (see *Gao et al., 2013*; *Jia and Gao, 2016*; *Jia and Gao, 2019*; *Jia et al., 2021a*). Besides the neotenic *Regalerpeton* as aforementioned, habitat preferences of these metamorphosed taxa and paleoecological disparity patterns of Cryptobranchoidea remain largely unexplored mainly due to yet established osteological indicators for ecology (see Discussion). The configuration of vomerine teeth has long been identified as useful

for the classification of living cryptobranchoids (*Zhao and Hu, 1984*) and was recently claimed to be ecologically informative (*Jia et al., 2021b*), but such statements have not received rigorous tests with inclusion of fossil taxa.

Our series of studies on living and fossil cryptobranchoids noticed that besides the vomerine teeth, the palate varies in shape and proportion, and could potentially serve as an indicator for paleoecological reconstruction (*Jiang et al., 2018*; *Jia et al., 2019*; *Jia et al., 2021b*; *Figure 1* and *Figure 1—figure supplements 1–23*). To test these hypotheses and to address the constraints underlying the morphological disparity of the palate, here we conducted a 2D landmark-based geometric morphometric analysis on the palate of all living and most aforementioned fossil genera of cryptobranchoids, stem, and other basal crown urodeles based primarily on micro-CT scanned specimens. We statistically investigated disparity patterns within the morphospace of the palate with respect to ecological preferences, life history strategies, and taxonomic affiliations. Based on a time-calibrated cladogram we established for fossil and living cryptobranchoids (*Jetz and Pyron, 2018*; *Jia et al., 2021a*), we further quantified the evolutionary rate of the palate and reconstructed the ancestral states for ecological preferences, life history strategies, palate shape, and vomerine tooth configurations of the respective last common ancestor of Panhynobia, Pancryptobrancha, Cryptobranchoidea, Urodela, and Caudata. We demonstrate that the palate is a reliable proxy in ecological reconstructions for early salamanders, and the morphospace of the palate is predominantly shaped by ecological constraints and also displays a stepwise evolutionary pattern.

## Results

In ventral view of the palate, the anteromedial fenestra is present between the vomer and the upper jaw in most early salamanders and is only absent in living cryptobranchids (*Figure 1—figure supplements 1–23*). The paired vomers medially articulate with each other in most taxa and posteriorly overlap to different extents, the anterior part of the cultriform process of the parasphenoid and/or the orbitosphenoid. The teeth are closely packed as a continuous tooth row positioned along the anterolateral periphery of the vomer in cryptobranchids but have diversified configurations in other taxa. The parasphenoid is a sword-like, azygous bony plate with its anterior part articulating dorsally with the orbitosphenoid and its posterior part flooring the otic capsule.

### Morphospace and shape disparity patterns of the palate in early salamanders

The palate is symmetric about the mid-sagittal plane of the skull with symmetric shape components accounting for 96.15% of the total shape variation in 70 specimens, and the left-right asymmetry accounting for the remaining 3.85%. The shape and the size of the palate with the latter represented by the centroid size (CS), are significantly correlated as revealed by the standard multivariate regression between log (CS) (independent variable) and symmetric shape components (dependent variable) across 70 specimens ($R^2$ = 9.3331%; p<0.001; $F$ = 6.9977; $Z$ = 3.918) and 34 species ($R^2$ = 13.276%; p<0.001; $F$ = 4.8985; $Z$ = 3.1691). However, when phylogenetic relationships of the 34 species were factored in, the association between size and shape of the palate is no longer significant as shown in the evolutionary allometry analysis (p=0.1583). Such inconsistency between standard and evolutionary allometry analyses is related to the fact that the CS of the palate has a strong phylogenetic signal (Blomberg's $K$ = 0.997, p=0.001, $Z$ = 4.1921) and, hence, the CS accounts for an even smaller amount of shape variations of the palate ($R^2$ = 4.494%; $F$ = 1.5059; $Z$ = 1.0141) when evolutionary history among species was counterbalanced in the evolutionary allometry analysis. To eliminate impacts from both asymmetry and allometry on the spatial patterns of the palate, residuals from the multivariate regression of symmetric shape components on log (CS) were retained for downstream statistical analyses.

To visualize the spatial patterns of the palate in the morphospace, on the basis of the size-corrected (allometry-free) 24-landmark dataset, we conducted a standard principal component analysis (PCA) across 70 specimens; and we also conducted three other types of PCA across 34 species with the time-calibrated cladogram and ancestral internal nodes projected into the morphospace (*Figure 2—figure supplements 1–3*), including: a phylomorphospace analysis (PA), phylogenetic principal component analysis (Phylo-PCA) and a phylogenetically-aligned components analysis

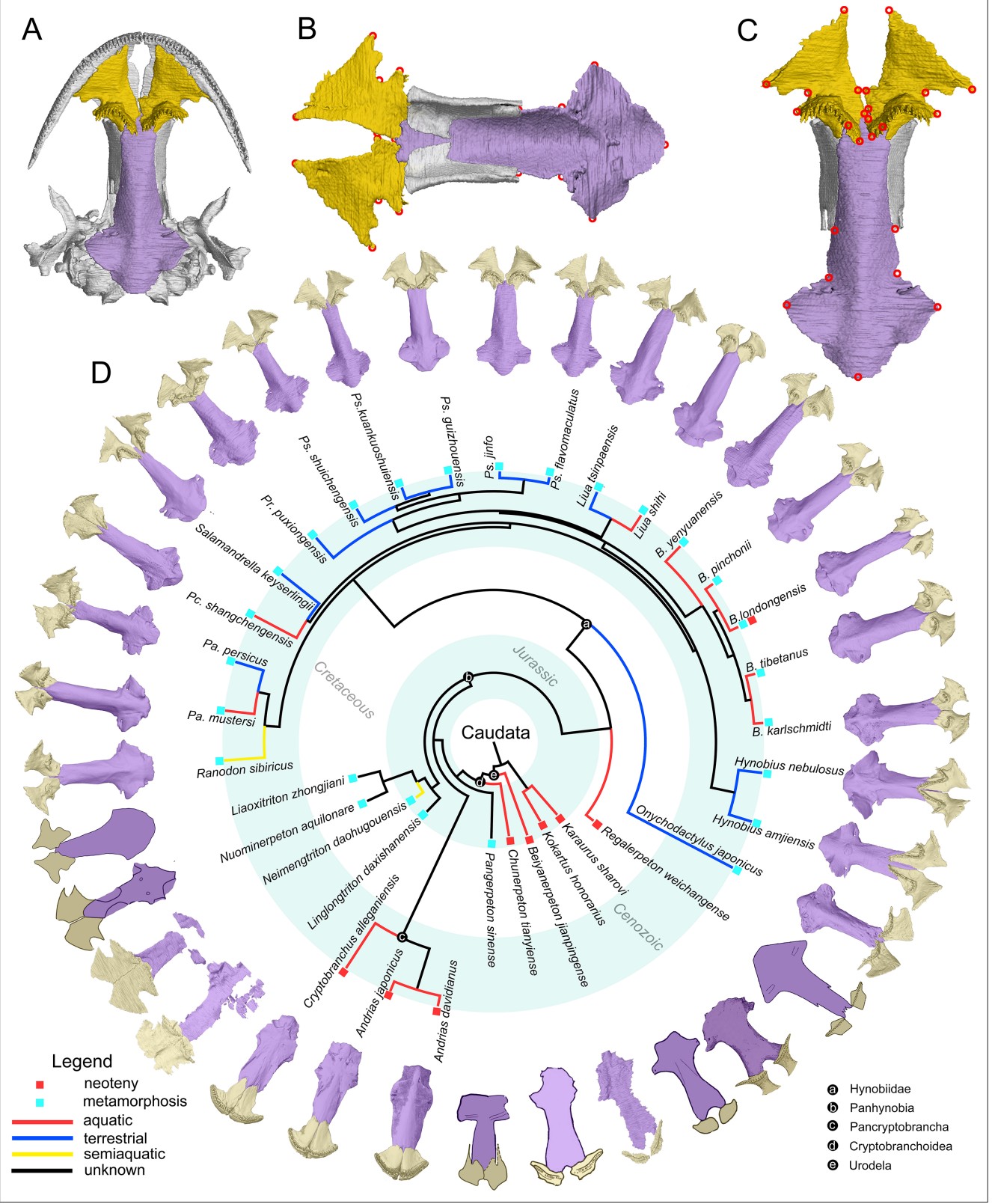

**Figure 1.** The palate and phylogenic relationships of early salamanders. (**A**) The vomer (gold) and parasphenoid (purple) of the palate in ventral view of the skull in living hynobiid *Pseudohynobius flavomaculatus*. (**B**) Dorsal view of the palate showing the articulation patterns with the paired orbitosphenoid (whitish). (**C**) Enlarged view of the palate in ventral view with red circles corresponding to the 24 landmarks used for the geometric

*Figure 1 continued on next page*

*Figure 1 continued*

morphometric analysis. (**D**) Palatal configurations of early salamanders in ventral view, with color-coded life history strategies (square block) and ecological preferences (line) plotted on the time-calibrated tree modified from *Jetz and Pyron, 2018* and *Jia et al., 2021a*.

The online version of this article includes the following figure supplement(s) for figure 1:

**Figure supplement 1.** CT rendering of the palate and orbitosphenoid of extant cryptobranchids in dorsal (first and third columns) and ventral (second and fourth columns) views.

**Figure supplement 2.** Images showing the ventral view of the right vomer (PIN 4357/13) of the Paleocene pancryptobranchan *Aviturus exsecratus*, with landmarks represented by red circles.

**Figure supplement 3.** CT rendering of the palate and orbitosphenoid of two species of living hynobiid *Batrachuperus* in dorsal (first and third columns) and ventral (second and fourth columns) views.

**Figure supplement 4.** CT rendering of the palate and orbitosphenoid of three species of living hynobiid *Batrachuperus* in dorsal (first and third columns) and ventral (second and fourth columns) views.

**Figure supplement 5.** CT rendering of the palate and orbitosphenoid of two species of living hynobiid *Hynobius* in dorsal (first and third columns) and ventral (second and fourth columns) views.

**Figure supplement 6.** CT rendering of the palate and orbitosphenoid of two species of living hynobiid *Liua* in dorsal (first and third columns) and ventral (second and fourth columns) views.

**Figure supplement 7.** CT rendering of the palate and orbitosphenoid of the living hynobiid *Onychodactylus japonicus* in dorsal (first and third columns) and ventral (second and fourth columns) views.

**Figure supplement 8.** CT rendering of the palate and orbitosphenoid of the living hynobiid *Pachyhynobius shangchengensis* in dorsal (first and third columns) and ventral (second and fourth columns) views.

**Figure supplement 9.** CT rendering of the palate and orbitosphenoid of two species of the living hynobiid *Paradactylodon* in dorsal (first and third columns) and ventral (second and fourth columns) views.

**Figure supplement 10.** CT rendering of the palate and orbitosphenoid of the living hynobiid *Protohynobius puxiongensis* in dorsal (first and third columns) and ventral (second and fourth columns) views.

**Figure supplement 11.** CT rendering of the palate and orbitosphenoid of three species of the living hynobiid *Pseudohynobius* in dorsal (first and third columns) and ventral (second and fourth columns) views.

**Figure supplement 12.** CT rendering of the palate and orbitosphenoid of two species of the living hynobiid *Pseudohynobius* in dorsal (first and third columns) and ventral (second and fourth columns) views.

**Figure supplement 13.** CT rendering of the palate and orbitosphenoid of the living hynobiid *Ranodon sibiricus* in dorsal (first and third columns) and ventral (second and fourth columns) views.

**Figure supplement 14.** CT rendering of the palate and orbitosphenoid of the living hynobiid *Salamandrella keyserlingii* in dorsal (first and third columns) and ventral (second and fourth columns) views.

**Figure supplement 15.** CT rendering of the palate of the Late Jurassic basal salamandroid *Beiyanerpeton jianpingense* in ventral view.

**Figure supplement 16.** CT rendering of the palate and orbitosphenoid of the Middle Jurassic stem hynobiid *Neimengtriton daohugouensis* in dorsal (first and third columns) and ventral (second and fourth columns) views.

**Figure supplement 17.** CT rendering of the palate and orbitosphenoid of the Late Jurassic stem hynobiid *Linglongtriton daxishanensis* in dorsal (first and third columns) and ventral (second and fourth columns) views.

**Figure supplement 18.** Line drawings of the palate and orbitosphenoid of the Early Cretaceous stem hynobiid *Liaoxitriton zhongjiani* in ventral view.

**Figure supplement 19.** Line drawings of the palate and orbitosphenoid of the Early Cretaceous stem hynobiid *Nuominerpeton aquilonare* in ventral view.

**Figure supplement 20.** Line drawings of the palate and orbitosphenoid of the Early Cretaceous stem hynobiid *Regalerpeton weichangense* in ventral view.

**Figure supplement 21.** Line drawings of the palate and orbitosphenoid of the Middle Jurassic basal cryptobranchoid *Chunerpeton tianyiense* in ventral view.

**Figure supplement 22.** Line drawings of the palate of the Late Jurassic basal cryptobranchoid *Pangerpeton sinense* in ventral view.

**Figure supplement 23.** Ventral view of the palate of two stem urodeles, the Late Jurassic *Karaurus sharovi* (upper row) and the Middle Jurassic *Kokartus honorarius* (lower row).

**Figure supplement 24.** The palate and phylogenic relationships of early salamanders with the inclusion of *Aviturus*.

**Figure supplement 25.** Configuration and superimposition of the 24 landmarks of the palate resulted from generalized Procrustes analyses across 70 specimens (**a**) and 34 species (**b**).

**Figure supplement 26.** Time calibrated cladograms used in this study with both terminal and internal taxa numerically labeled.

(PaCA). The first three PC axes in each of the four PCAs collectively measure up to about 70% of total shape variances (*Supplementary file 1A*). Within the phylomorphospace defined by principal components (PCs) 1–2 (*Figure 2B* and *Figure 2—figure supplement 1b*), aquatic and terrestrial living species of Cryptobranchoidea are generally located along the positive and negative interval of the PC 2 axis (21.35%), respectively. Most taxa are ecologically exclusive at the genus level, except that *Liua* and *Paradactylodon* are the only two living hynobiid genera with species occupying both the aquatic (*Liua shihi* and *Paradactylodon mustersi*) and terrestrial (*Liua tsinpaensis* and *Paradactylodon persicus*) zones. The living semiaquatic hynobiid *Ranodon sibiricus* occupies at a location intermediate between the aquatic and terrestrial zones. Shape changes of the palate relative to the mean shape of all terminal and internal taxa along the positive values of PC 2 (aquatic zone) involve an anteromedial extension of the vomer and therefore a reduction of the anteromedial fenestra, a shrinkage of the anterolateral and posterolateral borders of the vomer and the width of the cultriform process of the parasphenoid, and an anterior extension of the parasphenoid. In contrast, the negative values of PC 2 (terrestrial zone) characterize a shrinkage of the anteromedial border of the vomer and therefore a posterolaterally expanding anteromedial fenestra, a laterally widening retrochoanal process of the vomer and the cultriform process of the parasphenoid, and an anteroposteriorly shortening parasphenoid. On the other hand, basalmost crown urodeles (e.g. *Beiyanerpeton*, *Chunerpeton*, and *Pangerpeton*) and karaurids are largely separated from living taxa along the PC 1 axis (37.49%), with stem hynobiids widely scattered in the phylomorphospace and lying within either the aquatic (*Liaoxitriton*) or terrestrial (*Linglongtriton* and *Nuominerpeton*) zone. The only known aquatic stem hynobiid *Regalerpeton* is situated at a region of the aquatic zone occupied by other aquatic fossil taxa (e.g. *Beiyanerpeton* and *Chunerpeton*), whereas the semiaquatic stem hynobiid *Neimengtriton* occupies a location closer to terrestrial than either aquatic zone or the semiaquatic zone of extant hynobiids. From the largest to the smallest value of PC 1, both the vomer and the parasphenoid have an anteroposterior extension and the cultriform process of the parasphenoid changes from an anteriorly widened plate with an indented anterior edge into a bilaterally narrowed plate with a pointed anterior edge.

The evolutionary history among species has a moderate but significant contribution in the formation of the spatial patterns in the phylomorphospace (phylogenetic signal: observed $K_{mult}$ = 0.4154, p=0.001, $Z$ = 4.9856). When the multivariate shape data of the palate were maximally aligned with phylogenetic signal (*Figure 2C* and *Figure 2—figure supplement 1c*), fossil taxa can be roughly divided from living taxa along PaCA-C 1. Both crown and stem taxa of Panhynobia are more compactly clustered than that seen in the phylomorphospace created by PA, and are distinct from the Pancryptobrancha and the region occupied by karaurids, basal cryptobranchoids and salamandroids; but taxa in different types of ecological preference are mixed together. By contrast, when the phylogenetic signals of the palate were eliminated by the Phylo-PCA (*Figure 2D* and *Figure 2—figure supplement 1d*), the overall spatial patterns among species are essentially preserved, albeit slightly rotated clockwise, as observed in the phylomorphospace. In this phylogeny-free morphospace, the last common ancestors of Pancryptobrancha and of Hynobiidae lie within the aquatic zone and are tightly associated with living cryptobranchids and the stem urodele *Kokartus*, respectively. In contrast, the last common ancestors of Panhynobia, Cryptobranchoidea, Urodela, and Caudata lie within the terrestrial zone and are adjacent to three highly derived living hynobiids, respectively, *Pseudohynobius shuichengensis*, *Pseudohynobius jinfo,* and *Liua tsinpaensis*.

Our pairwise comparison (*Supplementary file 1B–E*) and phylogenetic Procrustes ANOVA reinforce that the shape of the palate is significantly different among groups of species that are classified by ecological preference ($R^2$ = 28.079%; p=0.001; F = 4.3740; Z = 3.4838) and life history strategy ($R^2$ = 6.797%; p=0.006; F = 3.1766; Z = 2.4673), but not by taxonomic affiliations, neither at the genus ($R^2$ = 68.711%; p=0.248; F = 1.2549; Z = 0.78492) nor family ($R^2$ = 12.858%; p=0.581; F = 0.8263; Z = –0.21234) level. With regard to the life history strategy, living cryptobranchids are separated from all Mesozoic neotenic taxa including *Beiyanerpeton*, *Chunerpeton*, karaurids, and *Regalerpeton* by the vast majority of the metamorphosed taxa, which occupy most of the morphospace of the palate (*Figure 2—figure supplement 4*). The single living neotenic hynobiid *B. londongensis* is situated between the two neotenic groups as aforementioned and lies alongside its metamorphosed conspecifics and those all collectively overlap with other aquatic hynobiids. When the seven-landmark-dataset for the right vomer was analyzed following the same procedure, the purported semiaquatic basal

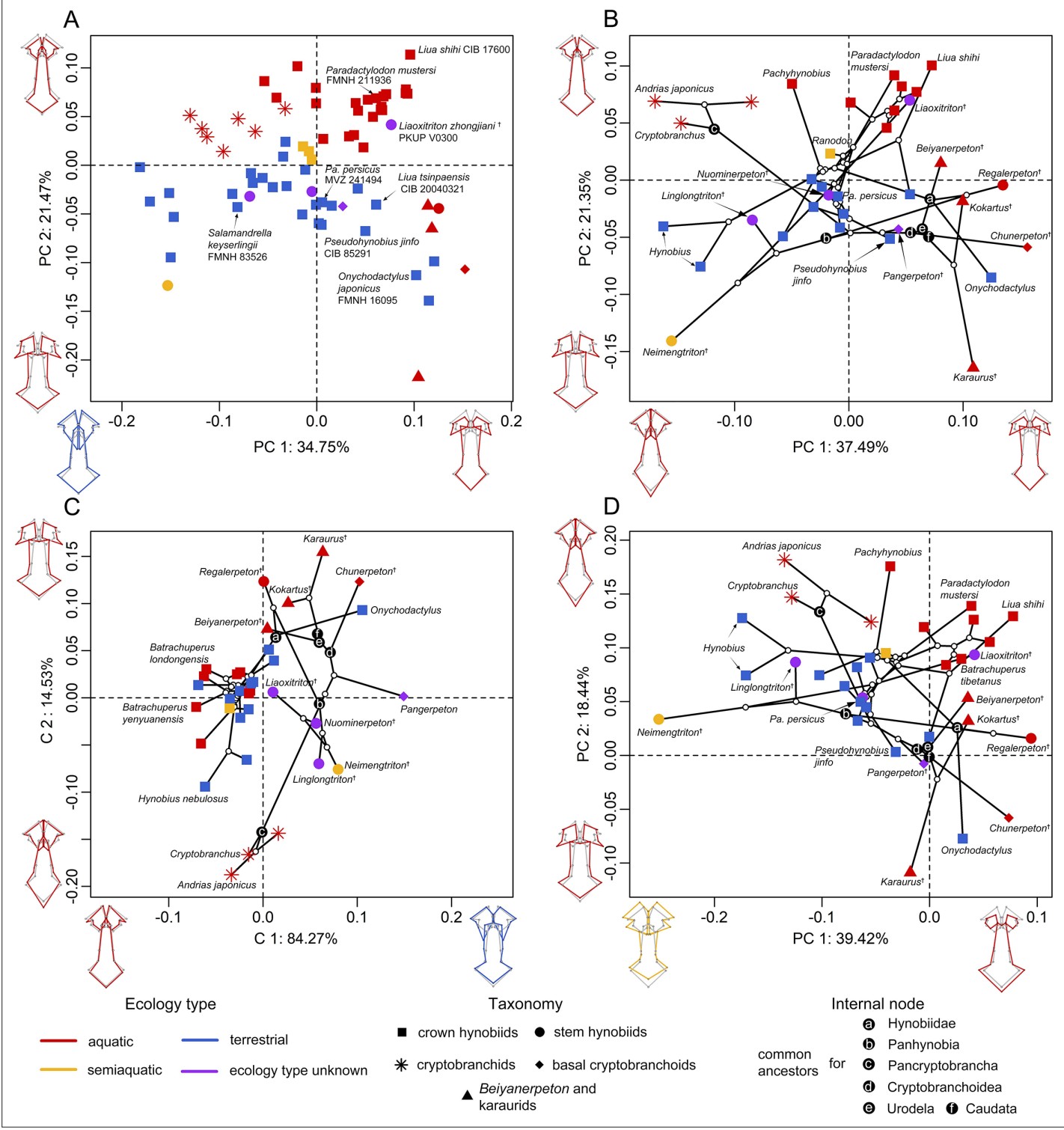

**Figure 2.** Spatial patterns of the palatal shape in the morphospace defined by the first two components generated from four principal component analyses (PCA). (**A**) Standard PCA across 70 specimens, (**B**) phylomorphospace analysis, (**C**) phylogenetically aligned components analysis, and (**D**) phylogenetic PCA across 34 species with ancestral states for internal nodes (open circles) and phylogenetic relationships (black lines) plotted in the morphospace. The color and shape of each point represent the ecological type and taxonomic affiliation, respectively. Extreme values of the palatal shape along both principal components (PCs) 1 and 2 are represented by wireframes color-coded to ecological types against the mean shape (gray) of both terminal and internal taxa.

The online version of this article includes the following figure supplement(s) for figure 2:

*Figure 2 continued on next page*

*Figure 2 continued*

**Figure supplement 1.** Spatial patterns of the palatal shape in the morphospace defined by the first two components generated from four principal component analyses (PCA).

**Figure supplement 2.** Spatial patterns of the palatal shape in the morphospace defined by principal components (PCs) 1 and 3 generated from 4 principal component analyses (PCA).

**Figure supplement 3.** Spatial patterns of the palatal shape in the morphospace defined by principal components (PCs) 2 and 3 generated from four principal component analyses (PCA).

**Figure supplement 4.** Spatial patterns of the palatal shape in the morphospace defined by the first two components generated from four principal component analyses (PCA).

**Figure supplement 5.** Spatial patterns of the shape of the right vomer in the morphospace defined by first two principal components (PCs) generated from four principal component analyses (PCA).

pancryptobranchan *Aviturus* is nested within the terrestrial zone along with living metamorphosed taxa (*Figure 2—figure supplement 5*).

## Evolutionary rates of the palate

The palate exhibits considerable differences in morphological disparity and evolutionary rates across regions represented by the 24-landmark points as calculated under a Brownian motion evolutionary model (*Figure 3*; *Supplementary file 1B–H*). The vomer evolves about two times faster than the parasphenoid, and such a pattern is also supported by the fact that the vomer has a higher phylogenetic signal ($K_{mult}$ = 0.6407) than that of the palate ($K_{mult}$ = 0.4154). The posterior border of the vomer and the medial-most part of the choanal notch are the fastest and the slowest evolving parts in the palate, respectively. In the parasphenoid, however, the highest evolutionary rates are concentrated on the anterior end, posterior end and the lateral alae, and the lowest evolutionary rates are located at the posterior contact between the cultriform process and the orbitosphenoid and the junction area where the cultriform process merges with the lateral alae.

Species in different ecological (p=0.002, $Z$ = 2.6284) and taxonomic (p=0.035, $Z$ = 1.7594) groups vary greatly in evolutionary rate of the palate (*Supplementary file 1H*). Namely, aquatic and terrestrial taxa are comparable with each other in the evolutionary rate of the palate, the vomer and the parasphenoid, and both of them evolve at a rate that is less than half of that in semiaquatic taxa championed by *Neimengtriton*. Stem hynobiids have the highest evolutionary rates, followed successively by basal cryptobranchoids, pancryptobranchans, crown hynobiids, and non-cryptobranchoid basal salamanders. Neotenic and metamorphosed taxa are not significantly different from each other in evolutionary rates (p=0.843, $Z$ = –1.1341).

## Non-shape covariates from the vomer

Similar to shape variables, allometry has a significant impact on the first 4 of the 5 continuous non-shape covariates of the palate (length ratios between parasphenoid and palate, and that between vomer and palate; width ratios between vomerine tooth row [VTR] and vomer, and between outer and inner branches of the VTR; and vomerine tooth number) when each of them is statistically regressed on the log (CS). We thus use residuals from the regression for boxplots to visualize the distribution patterns of the covariates in three ecological groups across 70 specimens, and use the Mann-Whitney U-test to analyze if the covariates are reliable ecological indicators (*Figure 3*). In line with shape changes revealed from PCA, aquatic and terrestrial species are significantly different in all of the five covariates, with aquatic taxa having proportionally a longer parasphenoid (median: ~0.84), a shorter vomer (median: ~0.28), a higher ratio in outer/inner branch of the VTR (median: ~1.6), a narrower VTR (except neotenic taxa; median: ~0.37) and fewer vomerine teeth (median: 7) than that of terrestrial taxa (i.e. ~0.78, ~0.33, ~0.58, ~0.64, and 16). Semiaquatic species cannot be confidently differentiated from aquatic or terrestrial taxa in any of the five covariates, but interestingly their parasphenoid length and VTR width resemble those in terrestrial taxa and their vomer length, outer/inner branch VTR width ratio and teeth numbers are more similar to aquatic taxa. The contingency table and Cochran-Mantel-Haenszel statistical test (*Figure 3—figure supplement 1* and *Supplementary file 1I*) show that the arrangement and position of VTR are both significantly correlated to ecological preference and life history strategy. The VTR in anterior and middle position of the vomer or being

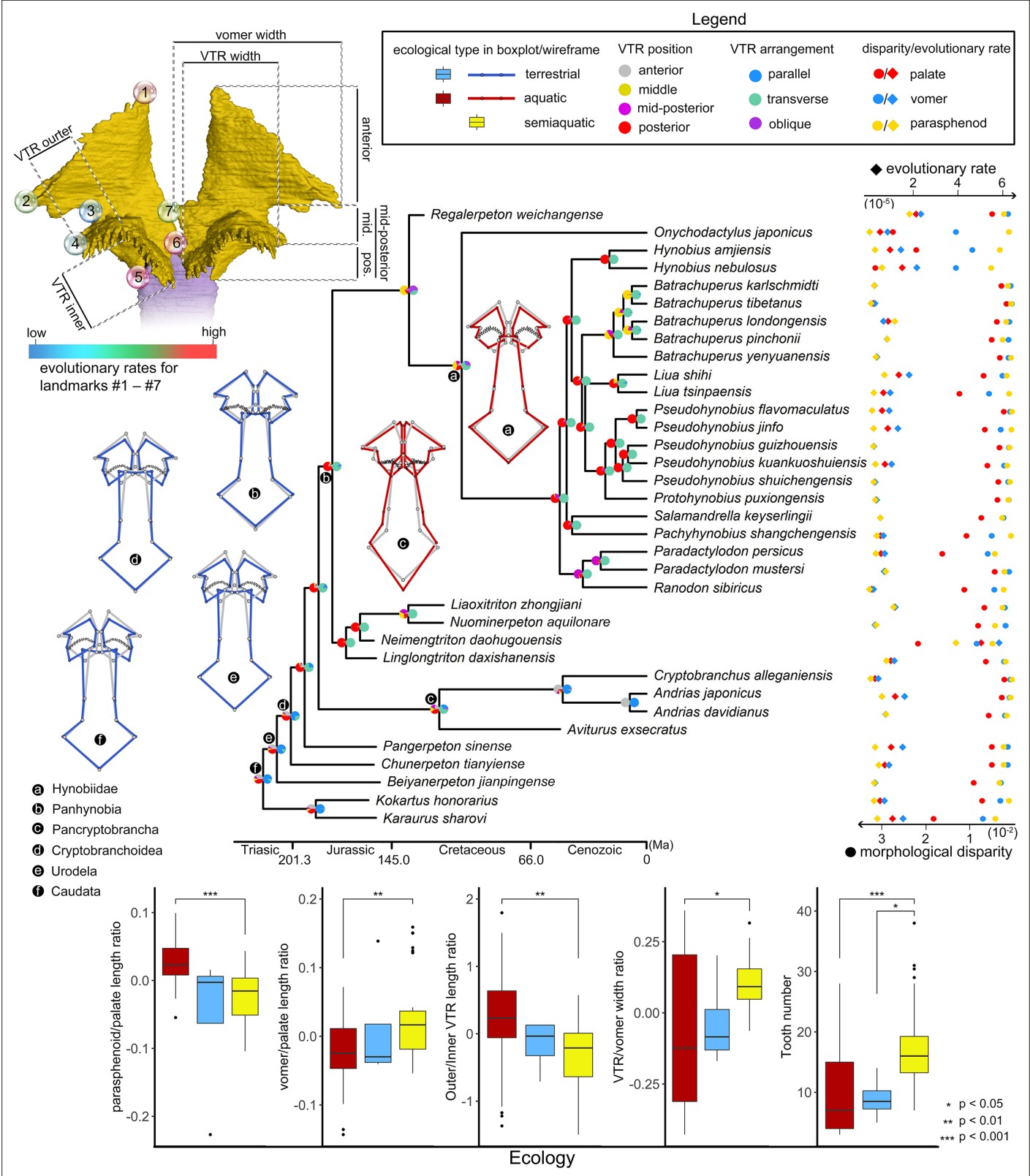

**Figure 3.** Evolutionary patterns of both the shape and non-shape covariates of the palate and their association with ecological disparity in early salamanders. Ancestral shape (wireframes color-coded to ecological types superimposed with mean shape [gray]) and vomerine tooth row (zigzag black lines) configurations are reconstructed for respective last common ancestors of Hynobiidae, Panhynobia, Pancryptobrancha, Cryptobranchoidea, Urodela, and Caudata. A complete list of evolutionary rates for each of the 24 landmarks, the vomer, parasphenoid, the palate and the continuous

*Figure 3 continued on next page*

Figure 3 continued

covariates of the vomer across 34 species is available in *Supplementary file 1GH and J*. The two pie charts at each internal node of the time-calibrated cladogram are likelihoods of the position (left) and arrangement (right) of the vomerine tooth row reconstructed in this study. Continuous covariates of the vomer were subjected to Mann-Whitney U test for their association with the three ecological groups with corresponding p values labeled above the boxplots.

The online version of this article includes the following figure supplement(s) for figure 3:

**Figure supplement 1.** Probability of the indicativeness of ecological preference (**a, b**) and life history strategy (**c, d**) from two discrete characters of the vomerine tooth row, the position (VTRP) and arrangement (VTRA) based on contingency table (*Supplementary file 1I*).

**Figure supplement 2.** Time-calibrated cladogram of 34 species showing the ancestral states of the life history strategy for internal nodes reconstructed by maximum likelihood.

arranged obliquely or parallel to the marginal tooth row are only present in aquatic taxa, with the VTR located anteriorly exclusively present in neotenic species. The VTR in the mid-posterior position of the vomer or when it is transversely arranged is only present in metamorphosed taxa with all kinds of ecological preferences. Species with VTR in the posterior position or transversely arranged are more likely to be terrestrial than aquatic or semiaquatic.

## Ancestral states of the palate

The ancestral states in both the shape and non-shape covariates of the palate were reconstructed for the last common ancestors of Hynobiidae, Panhynobia, Pancryptobrancha, Cryptobranchoidea, Urodela, and Caudata, respectively, using maximum likelihood (*Supplementary file 1J*). When compared to the mean shape of the palate, the configurations of the palate in the respective last common ancestors of Caudata, Urodela, and Cryptobranchoidea are characterized by a shortened parasphenoid with an anteriorly widened cultriform process and an anteriorly shortened but posteriorly widened vomer. This configuration coincides with our reconstructions of the respective last common ancestors of Caudata, Urodela, and Cryptobranchoidea is parallel to the marginal tooth row and is located in the posterior part of the vomer with the outer longer than the inner branch. The VTR retained a similar configuration, except for being transversely arranged in the last common ancestors of Panhynobia and Pancryptobrancha and then shifting to the middle part of the vomer in the last common ancestor of Hynobiidae.

## Discussion

Among modern amphibians, anurans, and caecilians undergo metamorphosis or direct development, and the adults of most species are terrestrial. In contrast, salamanders have their ecological preference decoupled from life history strategy, especially in metamorphosed taxa such as living and extinct hynobiids where postmetamorphosed adults live in water, on land or are semiaquatic (*Fei et al., 2006*; *Jia and Gao, 2016*; *Jia et al., 2021a*). The discrepancy in ecological preference between salamanders on one hand and anurans and caecilians on the other hand is unhelpful in understanding the evolutionary paleoecology in the early lissamphibians given that salamanders and anurans are sister-groups. Previous studies on fossil salamanders generally focus on the taxonomy, whereas the paleoecology and the evolutionary history of the ecological decoupling particularly in metamorphosed taxa had received insufficient attention, which is to some extent explained by the insufficient taphonomic analyses on fossil sites with salamander discoveries, such as the Daohugou fossil locality (*Wang et al., 2019*). The main obstacles are (1) sufficient and reliable ecological indicators in the bony skeleton have not yet been established for extant cryptobranchoids and early salamanders (*Xiong et al., 2016*; *Jia et al., 2021b*), and (2) soft anatomical structures (e.g. labial fold and caudal fin) and stomach contents that are ecologically informative, as commonly seen in lacustrine deposits of certain neotenic species (*Dong et al., 2011*), are rarely preserved in metamorphosed individuals. Here, we quantitively investigate the paleoecological turnover in the earliest known salamanders based on the shape and non-shape variables of the palate. Our results shed light on the interactions between morphology and paleoecology, and the underlying mechanisms governing ecomorphological diversity among early salamanders along the rise of modern amphibians.

As shown in the phylomorphospace, the shape of the palate in early salamanders is heavily impacted by convergence resulting from ecological disparity, as extant cryptobranchoids and fossil taxa in

different ecological preferences occupy distinctive zones in the morphospace. Such spatial patterns remain essentially unaltered after phylogenetic signals are eliminated by the Phylo-PCA, along the first axis of which aquatic and terrestrial taxa regardless of their phylogenetic closeness are separated from each other, confirming our hypothesis that the shape of the palate is a reliable ecological indicator in early salamanders as reported in living salamanders (*Fabre et al., 2020*). As shown above, non-shape covariates of the palate particularly with regard to the VTR are also ecologically informative. The palate in aquatic species is characterized by a reduction of the anteromedial fenestra, a shrinkage of the vomer laterally and anteroposteriorly with fewer vomerine teeth, a high ratio of the outer to the inner branch of the VTR, a narrow VTR, and an elongated parasphenoid. On the other hand, the palate in terrestrial species is characterized by a posterolateral expansion of the anteromedial fenestra, an anteroposteriorly elongated vomer and a widened retrochoanal process with more vomerine teeth, a low ratio of the outer to inner branch of the VTR, a widened VTR, and an anteroposteriorly shortened parasphenoid. The palate in extant semiaquatic taxa is transitional in both the morphospace and many non-shape covariates between aquatic and terrestrial taxa as mentioned above, whereas the palate of the only known fossil semiaquatic salamander, *Neimengtriton* (Middle Jurassic) is configured more like that of terrestrial taxa as argued in the original study (*Jia et al., 2021a*), indicating that *Neimengtriton* is probably more adapted to terrestrial environments despite being semiaquatic. The respective last common ancestors for Caudata, Urodela, Cryptobranchoidea, and Panhynobia, the basal cryptobranchoid *Pangerpeton* (Late Jurassic) and the stem hynobiids *Linglongtriton* (Late Jurassic) and *Nuominerpeton* (Early Cretaceous) are terrestrial, whereas other stem hynobiids *Liaoxitriton* and *Regalerpeton* (Early Cretaceous) and the respective last common ancestors for Hynobiidae and Pancryptobrancha are aquatic, demonstrating that ecological shifts occurred frequently in the evolution of early salamanders. The Paleocene pancryptobranchan *Aviturus* is likely metamorphosed and terrestrial based on the shape and non-shape variables of the vomer, but this hypothesis awaits to be tested by the discovery of a more completely preserved palate.

Morphological adaptations in the palate and perhaps in other cranial features (e.g. hyobranchial apparatus; see *Jia et al., 2021b*) to aquatic and terrestrial living settings are likely constrained by feeding mechanisms (*Regal, 1966*; *Reilly and Lauder, 1990*). In extant cryptobranchoids, terrestrial species grasp prey items with their jaws and/or by their tongue protruding from the mouth; the captured prey is then transported from the snout into the esophagus with the assistance of several cyclical movements of tongue and hyobranchial apparatus that press and reposition the prey against the palate (*Deban and Wake, 2000*; *Wake and Deban, 2000*). Such feeding mechanisms in terrestrial cryptobranchoids demand a sticky tongue moisturized by the intermaxillary or internasal gland that is housed above an expanded anteromedial fenestra, a wide snout contributed by lateral expansions of the vomer, many vomerine teeth to efficiently hold the prey in place, and a shortened parasphenoid to reduce the distance of intraoral transportation. The posteriorly elongated inner branch of the VTR as represented by derived terrestrial hynobiids *Hynobius* and *Salamandrella* would serve as toothed surfaces that can further facilitate tongue manipulations and transportations of small prey items posteriorly as in more sophisticated terrestrial feeders (e.g. salamandrids and plethodontids; *Regal, 1966*). By contrast, aquatic salamander species capture their prey primarily by suction feeding in which the prey is carried into the biting range of the jaws by water currents created by retraction and expansion of the buccal walls, and giant salamanders can even perform asymmetrical strikes in water via unilateral jaw and hyobranchial movements (*Gillis and Lauder, 1994*). Intraoral transport of the prey is more efficient in aquatic than in terrestrial species; and it typically involves repeating the same procedure as in the initial suction strike and is occasionally assisted by tongue manipulations against the palate (*Regal, 1966*). The limited usage of the tongue in aquatic prey capture diminishes the need for moisturization from the intermaxillary or internasal gland which, in turn, may lead to reduction and closure of the anteromedial fenestra as seen in aquatic species. An anteroposterior expansion of the parasphenoid increases buccal volume, and the shrinkage of the vomer along both the mediolateral and anteroposterior dimensions reduces snout size; both features act in concert to boost the success of feeding attempts by creating a high negative buccal pressure. A narrow VTR with few teeth, most of which are located on the outer branch, likely serves to reduce hindering the influx of water and prey.

Whichever mode of prey capture and intraoral transportation cryptobranchoids perform, the palate undergoes a similar mechanical stress distribution pattern as found in species with different types of ecological preference and life history strategy (*Fortuny et al., 2015*: *Andrias davidianus*;

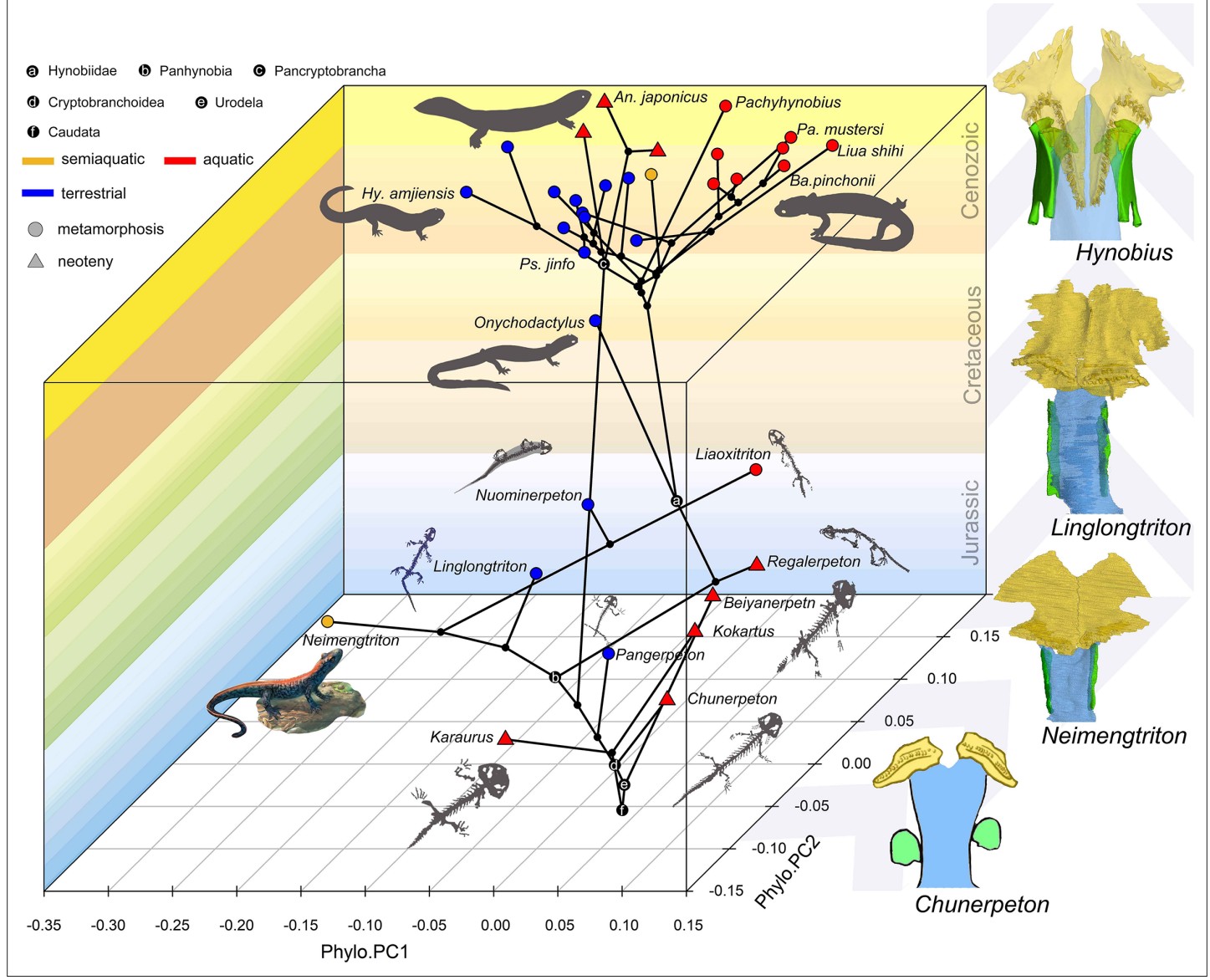

**Figure 4.** Spatial-temporal patterns of phenotypic diversities of the palate and their associations with ecological preference and life history strategy in early salamanders and the stepwise evolutionary patterns at the sutural area of vomer, parasphenoid, and orbitosphenoid. The morphospace of the palate is defined within a space formed by geological time scale (Z-axis) and principal components (PCs) 1 (X-axis) and 2 (Y-axis) derived from phylogenetic principal component analysis across 34 species. All silhouettes and images of salamander species are original.

*Zhou et al., 2017*: *Salamandrella keyserlingii*). Such findings reinforce that alternative morphological configurations of certain features in the palate cope well with similar functional demands, but do not necessarily indicate that other feeding-related features of the palate and other regions of the cranium would also exhibit such extensive convergence. For example, the ceratohyal has been argued to be ecologically informative in extant cryptobranchids, as the ceratohyal is consistently ossified at its posterior end in aquatic species but remains cartilaginous in terrestrial taxa (*Xiong et al., 2016*; *Jia et al., 2021b*); however, the ceratohyal remains cartilaginous in all known early fossil stem and crown urodeles, and ossifications of the ceratohyal must represent a derived feature independently evolved in crown cryptobranchids (*Jia et al., 2021b*). Interestingly, the palate of extant cryptobranchids as shown in biomechanical analyses (*Fortuny et al., 2015*; *Zhou et al., 2017*) has the highest stress level concentrated in the most posterior part of the vomer and its sutural area with the parasphenoid and orbitosphenoid, which surprisingly correspond to the place with the highest evolutionary rate in the palate as revealed by our study. Here, we recognize a clear stepwise evolutionary pattern at the

vomer-parasphenoid-orbitosphenoid sutural area in early salamanders (*Figure 4*). In stem (*Karaurus* and *Kokartus*) and many basal crown (*Beiyanerpeton, Chunerpeton, Jeholotriton, Neimengtriton, Pangerpeton,* and *Qinglongtriton*) urodeles from the Jurassic and the reconstructed respective last common ancestors for Panhynobia, Cryptobranchoidea, Urodela, and Caudata, the vomer has a limited contact with the parasphenoid, the cultriform process of the parasphenoid is anteriorly expanded bilaterally, and the orbitosphenoid has no anteroventral processes (see below) or in the case of *Qinglongtriton* remains completely cartilaginous. In three stem hynobiids (*Liaoxitriton, Nuominerpeton,* and *Regalerpeton*) and the reconstructed ancestors for Hynobiidae and Pancryptobrancha from the Cretaceous and another stem hynobiid *Linglongtriton* from the Late Jurassic, the vomer has an increased contact with the parasphenoid, the cultriform process of the parasphenoid is anteriorly constricted, and the orbitosphenoid lacks the anteroventral process. In living cryptobranchoids, the vomer is in more extensive contact with the parasphenoid and orbitosphenoid (*Trueb, 1993*: *Cryptobranchus*; *Kuzmin and Thiesmeier, 2001*: *Ranodon*; *Jiang et al., 2018*: *Batrachuperus*; *Jia et al., 2021b*: *Pseudohynobius*), the cultriform process of the parasphenoid is anteriorly constricted bilaterally, and the orbitosphenoid has a thick anteroventral process ossified from neighboring cartilages in the nasal capsule that projects medially and firmly overlaps the cultriform process of the parasphenoid. If similar biomechanical stress patterns existed in early salamanders, the stress and strain applying to the vomer/parasphenoid sutural area would be equally offset by either a wide and thin cultriform process of the parasphenoid with limited support from the vomer and the absence of anteroventral process of the orbitosphenoid as seen in Jurassic taxa, or a narrow cultriform process thickened by more extensive overlapping with the vomer as seen in Cretaceous and living taxa, and by ossification of the anteroventral process of the orbitosphenoid as seen solely in living taxa.

It is worth emphasizing that the phenotypic diversity of the palate is variously affected by neoteny and metamorphosis. Neoteny is notable in producing convergent characters through truncation of normal developmental trajectories and its indirect role associated with constraints from the aquatic environment (*Wiens et al., 2005*). Most early stem and crown urodeles are neotenic (e.g. *Chunerpeton* and *Kokartus*) and their phenotypic diversity of the palate is truly confined to a restricted area in the spatial-temporal morphospace. However, extant neotenic cryptobranchids and the hynobiid *B. londongensis* bear an increased phenotypic diversity of the palate as compared to fossil neotenic taxa, and are separated from one another within the morphospace with the neotenic *B. londongensis* even overlapping with its metamorphosed conspecifics, indicating that constraints imposed by ecology have more influence than neoteny in the morphogenesis of the palate. The ecomorphospace of the palate is indeed greatly and rapidly expanded by metamorphosed taxa represented by basal cryptobranchoids and most stem hynobiids as these taxa have the highest evolutionary rate in the palate (*Supplementary file 1H*). The last common ancestors of Caudata, Urodela, and Cryptobranchoidea are reconstructed to be metamorphosed and terrestrial as evidenced by the shape and non-shape variables of the palate. Disparities of ecological preference among metamorphosed taxa must have taken place before the Middle Jurassic (Bathonian), because one of the oldest known crown urodeles, *Neimengtriton*, is metamorphosed and semiaquatic as evidenced by the presence of a low but pliable dorsal caudal fin at adult stage. Most of the current scope in the phenotypic diversity of the palate possessed by modern cryptobranchoids were achieved by Early Cretaceous. Our results rigorously show that the shape of the palate and many non-shape covariates particularly associated with vomerine teeth are reliable ecological indicators for paleoecology of early salamanders, and we demonstrate that metamorphosis with biphasic ecological preference (aquatic larvae + terrestrial adults) is not only the ancestral lifestyle in salamanders but also significant for the rise and diversification of modern amphibians.

## Materials and methods
### Experimental design, specimens, and palate

Our study includes 60 wet specimens (preserved in formalin) that represent 25 living species (29% of Hynobiidae and 75% of Cryptobranchidae) in all 12 living genera of Cryptobranchoidea (*Andrias, Batrachuperus, Cryptobranchus, Hynobius, Liua, Onychodactylus, Pachyhynobius, Paradactylodon, Protohynobius, Pseudohynobius, Ranodon,* and *Salamandrella*), and one fossil skeleton for each of five

stem hynobiids (*Liaoxitriton zhongjiani*, *Linglongtriton daxishanensis*, *Neimengtriton daohugouensis*, *Nuominerpeton aquilonare*, and *Regalerpeton weichangense*), one basal pancryptobranchan (*Aviturus exsecratus*), two basal cryptobranchoids (*Chunerpeton tianyiense* and *Pangerpeton sinense*), one basal salamandroid (*Beiyanerpeton jianpingense*) and two stem urodeles (*Karaurus sharovi* and *Kokartus honorarius*; Dataset in Dryad). Sirenids were not included into this study for the following four reasons: (1) their palate is extremely specialized and is patterned with an enlarged palatine that is absent in most other salamanders and patches of teeth densely arranged on palatine and vomer; (2) sirenids are the only herbivory salamanders with a complex three-dimensional chewing behavior (**Schwarz et al., 2020**), and their specially configured palate may receive biomechanical patterns different from other salamanders; (3) the palate is incompletely preserved in the earliest known sirenid taxon *Habrosaurus* from the latest Early Cretaceous (**Gardner, 2003**); and (4) the existence of a ~90 Ma fossil gap between *Habrosaurus* and the earliest known salamandroids (*Beiyanerpeton* and *Qinglongtriton*) greatly impede our understanding of their early evolution of the palate. To keep the gender of species names consistent with that of genus names as per ICZN codes, we replaced the feminine/masculine species ending ('-is') by corresponding neuter forms ('-e') for genus names (e.g. *Nuominerpeton*) ending in the neuter noun 'herpeton' or 'ἑρπετόν' in Greek as suggested in **Rong et al., 2021**. In this study, each species is represented by one to three specimens, except for the only facultatively neotenic cryptobranchoid *B. londongensis* where both neotenic and metamorphosed populations are each represented by three specimens. Most of the specimens are accessioned in the following seven institutional collections: Capital Normal University (CNU), Beijing, China; Chengdu Institute of Biology (CIB), Chengdu, Sichuan Province, China; Field Museum of Natural History (FMNH), Chicago, USA; Liupanshui Normal University (HNUL), Liupanshui, Guizhou Province, China; Peking University of Paleontological Collections (PKUP), Beijing, China; Zhejiang Museum of Natural History (ZMNH), Hangzhou, Zhejiang Province, China; and Zunyi Medical University (ZMU), Zunyi, Guizhou Province, China. For specimens that we were not able to examine firsthand, we used publicly available micro-CT scan data of 12 living cryptobranchoid specimens (Dataset in Dryad) from the MorphoSource platform (MorphoSource.org) and published images for five fossil taxa, including *Aviturus* (**Vasilyan and Böhme, 2012**: Figure 3), *Chunerpeton* (**Gao and Shubin, 2003**: Figure 1), *Karaurus* (**Ivachnenko, 1978**: Figure 1), *Kokartus* (**Skutschas and Martin, 2011**: Figure 9), and *Regalerpeton* (**Rong, 2018**: Figure 5).

Besides the CT scan data obtained from MorphoSource, four fossil and all living specimens were micro-CT scanned using the following three high-resolution CT scanners: a Nikon XT H 320 LC scanner in the Industrial Micro-CT Laboratory at China University of Geosciences (Beijing); a GE Phoenix v/tome/x 240kv/180kv scanner in the PaleoCT Lab at The University of Chicago; and a Quantum GX micro-CT Imaging System (PerkinElmer, Waltham, USA) at CIB (Dataset in Dryad). No filter was used because beam hardening artifacts were not encountered during CT scanning. The voxel size of the volume files generated from CT scans ranges between 14.52–87.32 µm (see detailed parameters in Dataset in Dryad). File processing including segmentation and rendering were accomplished by VG Studio Max (version 2.2; Volume Graphics, Heidelberg, Germany), and images in jpeg format showing the ventral view of the skull were exported for landmark acquisition after digital removal of the mandible and the hyobranchium from the virtual models.

The palate in adult specimens of cryptobranchoids consists of the partes palatinae of the premaxilla and maxilla, paired vomers and a single median parasphenoid. An independently ossified palatine is present in few fossil taxa and is absent in the remaining, and is therefore not included in this study as a compromise between taxa sampling and landmarks collection. The pars palatina of both the premaxilla and maxilla is a narrow bony ledge and invariably contributes to a small portion of the palate by posteriorly articulating with the vomer, and hence is not considered in this study (**Figure 1A**). Both the vomer and parasphenoid are dorsoventrally flattened bony plates and are homologous across species studied here, and thus are ideal for landmark collections for both living and fossil specimens, considering that the palate in all available fossil cryptobranchoids is dorsoventrally preserved. All anatomical terms used here follow **Trueb, 1993** unless otherwise stated.

## Cladograms

For multivariate phylogenetic comparative analyses (evolutionary allometry analysis, phylogenetic signal analysis, Phylo-PCA, PA, phylogenetic alignment component analysis, and phylogenetic

ANOVAs; see below), a maximum clade credibility tree (MCC; a target tree with maximum sum of posterior probabilities on its internal nodes) with 24 living taxa in our dataset was created by the software TreeAnnotator (version 1.10.4; *Drummond et al., 2012*) based on 10,000 random, time calibrated trees from a recent Bayesian posterior distribution analysis of living amphibians (*Jetz and Pyron, 2018*). To maximally match the terminal taxa in the MCC tree with our dataset, the living hynobiid *Batrachuperus taibaiensis* was eliminated from multivariate phylogenetic comparative analyses, considering that *B. taibaiensis* was recently synonymized to *Batrachuperus tibetanus* (*Fei and Ye, 2016*). The MCC tree was configured in Newick format and was read and visualized in the package 'ape' (version 5.5; *Paradis and Schliep, 2019*) for the software R (version 4.0.5; *R Development Core Team, 2021*). The remaining fossil taxa in our dataset with age ranges determined from published literature (Dataset in Dryad) and the Paleobiology Database (https://www.paleobiodb.org) were incorporated into this MCC tree based on a recent cladogram (*Jia et al., 2021a*: Figure 6) we constructed for living and fossil cryptobranchoids (*Figure 1D*). Adding fossil taxa increases the accuracy for multivariate phylogenetic comparative analyses and ancestral state reconstruction (see below; *Soul and Wright, 2021*), but will inevitably bring zero-length branches when the age of internal nodes was considered equal to that of its immediate oldest descendant, resulting in difficulties in multivariate phylogenetic comparative analyses. To circumvent this problem, any zero-length branch in the cladogram was treated to have a same branch length with its first none zero-length ancestral branch by using the 'equal' dating method in the 'DatePhylo' function from the R package 'strap' (version 1.4; *Bell et al., 2015*). We followed the suggestion from the 'strap' tutorial by setting the root length as 60 Ma by using the age difference between the oldest taxon in the cladogram (*Kokartus*; ~170 Ma) and the first older outgroup taxon known to date (*Triassurus*; ~230 Ma). Considering that the palate of *Aviturus* is only represented by the right vomer, the cladogram (*Figure 1—figure supplement 24*) including *Aviturus* was only used for the phylogenetic signal analysis and principal component analysis (PCA; see below) of the right vomer; whereas the cladogram without *Aviturus* was mapped onto the morphospace for a number of PCA and evolutionary rate analysis.

## Data acquisition, geometric morphometrics, and symmetry and asymmetry

The geometry of the paired vomers and parasphenoid is represented on the ventral view of the palate by 24 type I (intersections of biological structures) and type II (maximum of curvature) 2D landmark points (*Figure 1B*; *Bookstein, 1991*; *Slice et al., 1996*) digitized by using tpsUtil (64 bit; version 1.76; *Rohlf, 2015*) and tpsDig2 (version 2.31; *Rohlf, 2015*) based on images of 70 specimens (the 24-raw-landmark dataset thereafter; see above; Dataset in Dryad). Most specimens have completely-preserved palate except that the Paleocene *A. exsecratus* is known only by a right vomer (*Vasilyan and Böhme, 2012*: *Figure 3*). In order to investigate the paleobiology of *A. exsecratus*, we created a separate dataset with seven corresponding landmarks (#1–#7) for the right vomer across all 71 specimens (termed as the seven-raw-landmark dataset thereafter; Dataset in Dryad). To reduce measurement errors, specimens were arranged alphabetically by the name of the image files, then landmarks in each specimen were digitized by the same author (J.J.) following the same order (landmark 1–24; *Supplementary file 1K*). Procrustes coordinates (24-Procrustes-landmark dataset and seven-Procrustes-landmark dataset) for the geometry of the palate were obtained in the software MorphoJ (version 1.07 a; Windows platform; *Klingenberg, 2011*) after superimposing the raw landmark coordinates of all specimens through the Generalized Procrustes Analysis (GPA; *Bookstein, 1991*; *Figure 1—figure supplement 25*), which serves to minimize non-shape variations (size, location, and orientation) among specimens by rescaling, repositioning and rotating raw landmark configurations. After superimposition, centroid size of the palate for each specimen is calculated as square root of the sum of squared distances of all Procrustes landmarks of the palate from their corresponding centroid. The mean shape of specimen/species are calculated as the consensus from the superimposed landmark coordinates. Procrustes variances/distance between specimens are calculated as the square root of the sum of the squared distances between corresponding landmarks (*Klingenberg, 2016*). The mean value for each species (the species mean thereafter) in the composite cladogram was obtained by averaging Procrustes coordinates and centroid sizes when there are more than one representative specimens.

The geometry of the palate shows object symmetry (*Mardia et al., 2000*) with landmarks #15 and #20 lying in the midsagittal axis of the skull and the remaining 11 pair landmarks being generally

symmetric about the mid-sagittal plane (*Figure 1A*). To calculate respective contributions to the total amount of shape variation from symmetric and asymmetric shape components, we used the function 'C1v' in the software R to create a new double 24-raw-landmark dataset that contains all original raw landmark coordinates and their mirrored copy reflected about the midsagittal axis and relabeled to match the original landmark numbers (*Savriama, 2018*). This doubled 24-raw-landmark dataset was then imported into MorphoJ for superimposition, covariation matrix buildup and PCA. Considering that the symmetric and asymmetric components of shape variations occupy complementary subspaces in the shape tangent space defined by the doubled datasets for landmark configurations with object symmetry (*Mardia et al., 2000*; *Klingenberg et al., 2002*), the relative amount of contributions to shape changes from symmetric and asymmetric components equals to the sum of the eigenvalues of their corresponding PCs derived from the PCA. This procedure does not apply to the seven-landmark dataset because the right vomer has no object symmetry.

## Allometric analysis

To verify if allometry, covariation of shape with size, plays a role in shape variations, we first performed a regular multivariate regression (*Monteiro, 1999*; *Klingenberg, 2016*) of the symmetric shape components (dependent variables) from the 24-Procrustes-landmark dataset on the log-transformed centroid size (log [CS]; independent variable) across all 70 specimens and the 34 species by using the 'procD.lm' function in the 'geomorph' R package (version 4.0.0; *Adams et al., 2021*; *Collyer and Adams, 2021*). Then we conducted a permutation test of 10,000 iterations using the same function to test the significance of these two regular regressions, respectively, considering that our sample size (n=70 for specimens; n=34 for species) is not considerably larger than the number of variables (n=49) for the 24-Procrustes-landmark dataset. Statistical results from these regular regression analyses may be inaccurate given that variations (or residuals) in the shape components among species are correlated with their evolutionary relationships ('phylogenetic non-independence' in *Felsenstein, 1985*). To account for impacts from the phylogeny, we conducted a phylogenetic regression (or evolutionary allometry analysis) of the shape components on log (CS) across species by using the function 'procD.pgls' of 'geomorph,' which is a distance-based phylogenetic generalized least squares method (*Adams, 2014a*) that uses our composite cladogram (without *Aviturus*) to remove phylogenetic covariances among species under a Brownian motion model of evolution.

To remove impacts from allometric shape components, shape variables (Procrustes-landmark coordinates in shape tangent space) were transformed by using residuals from the regular multivariate regression of shape on log (CS) across 70 specimens and 34 species, respectively, by using the function 'procD.lm' in 'geomorph' (*Klingenberg, 2016*). Non-allometric portion of the symmetric shape components are used for downstream statistical analyses.

## Phylogenetic signal

The phylogenetic signal is termed based on a pattern that closely related species tend to have similar character values stemmed from their shared evolutionary history (*Felsenstein, 1985*; *Blomberg et al., 2003*; *Adams, 2014b*). Whether the shape of the palate bears any phylogenetic signal was tested by using the $K_{mult}$ method in the function 'physignal' of 'geomorph,' which uses the composite cladogram to account for phylogenetic non-independence under a Brownian motion model of evolutionary divergence, and permutates shape components of terminal taxa against the composite cladogram (without *Aviturus*). A phylogenetic signal analysis was also conducted for the right vomer (non-allometric seven-Procrustes-landmark dataset) using the composite cladogram including *Aviturus*. The resulting $K_{mult}$ value for each of the two analyses measures the phylogenetic signal as a ratio of observed to expected shape variation obtained with and without considering phylogenetic non-independence (*Adams and Collyer, 2019*). When the $K_{mult}$ value is larger than and equals to 1, closely related taxa resemble each other phenotypically more than expected under Brownian model, whereas when the value is smaller than 1, closely related taxa are less similar to one another phenotypically than expected under Brownian model (*Adams, 2014b*).

## Morphological disparity and ecological indication of the palate

Taxonomically, the five groups considered here are Pancryptobrancha, Hynobiidae, stem hynobiids, basal cryptobranchoids and non-cryptobranchoid early salamanders. The Paleocene *Aviturus* was

classified into the clade Pancryptobrancha (*Gubin, 1991*; *Vasilyan and Böhme, 2012*). The Late Jurassic *Beiyanerpeton* represents a basal member of Salamandroidea (*Gao et al., 2013*) and the two stem urodeles *Karaurus* and *Kokartus* were classified into the family Karauridae (*Skutschas and Martin, 2011*). Following *Jia et al., 2021a*, we apply the name Hynobiidae to the crown group of Panhynobia, and we consider crown and stem taxa of Panhynobia separately. As mentioned above, living hynobiids are mainly metamorphosed because individuals typically go through a short period of development during which larval features (e.g. external gills, gill slits) are resorbed (e.g. *Kuzmin and Thiesmeier, 2001*; *Kami, 2004*; *Fei et al., 2006*; *Poyarkov et al., 2012*), and *B. londongensis* represents the single facultatively neotenic hynobiid species; living cryptobranchids are generally referred to as neotenic or partially metamorphosed as their individuals live in water in the adult stage and retain a few paedomorphic features (*Deban and Wake, 2000*), except that the life history strategy of *Aviturus* remains unknown (*Vasilyan and Böhme, 2012*; but see *Skutschas et al., 2018*). It is noteworthy that many fossil taxa investigated here (e.g. *Chunerpeton*) bear more apparent neotenic features (e.g. external gills, gill rakers) than living cryptobranchids; however, we did not differentiate subgroups within neotenic taxa due to our sampling scope. Living habitats of cryptobranchoids in the adult stage outside the breeding season have been broadly classified into three types, namely aquatic, semiaquatic, and terrestrial (*Fei et al., 2006*; *Rong, 2018*; *Fei and Ye, 2016*; *AmphibiaWeb, 2021*). In order to understand if factors like taxonomic affiliations at the genus and family level, life history strategy and living habitat would contribute to morphological disparities of the palate, we conducted phylogenetic Procrustes ANOVAs for the non-allometric symmetric shape components of the 24-Procrustes-landmark dataset using the function 'procD.pgls' of 'geomorph.' Pair-wise comparisons for morphological disparity (Procrustes variance) for the palate, vomer, parasphenoid (*Supplementary file 1B*) and each landmark point (*Supplementary file 1C–E*) across each of the aforementioned category were performed using 'morphol.disparity' in 'geomorph.'

## Principal component analyses

To visualize the patterns of shape changes in the morphospace of palate and to investigate the mechanisms framing up the disparity patterns, we conducted four types of PCA by using the function 'gm.prcomp' in 'geomorph': a standard PCA, a phylogenetically aligned component analysis (PaCA; *Collyer et al., 2020*), a phylomorphospace analysis (PA; *Rohlf, 2002*), and a phylogenetic principal component analysis (Phylo-PCA; *Revell, 2009*). The standard PCA is based on a covariance matrix of the non-allometric symmetric shape components across all 70 specimens, and the resulting first few PCs (eigenvalues) reveal predominant trends in shape disparity patterns.

The other three PCA are based on the composite cladogram (without *Aviturus*) and a covariance matrix of the non-allometric symmetric shape components across all 34 species, with the ancestral PC values (eigenvalue) for internal nodes (*Figure 1—figure supplement 26*) estimated by using the Brownian motion model of evolution. The resulting first few PCs derived from the PA reveal predominant trends in shape disparity patterns with the phylogeny superimposed in the morphospace. On the other hand, the first PC derived from PaCA reveal shape disparity patterns mostly associated with phylogenetic signal, and the first few PCs from Phylo-PCA are phylogenetic independent and therefore reveal factors (e.g. ecological preferences) other than phylogenetic signal that account for disparity patterns of the palate (see *Collyer et al., 2020*). A same analysis strategy was conducted for the non-allometric seven-Procrustes-landmark dataset to address the paleobiology of *Aviturus* (*Figure 2—figure supplement 5*). Visualizations of the results were achieved by using the functions 'plot.gm.prcomp,' 'make_ggplot' in 'geomorph,' and 'ggplot' in the package 'ggplot2' (version 3.3.5; *Wickham, 2016*).

## Covariates of the palate

Configurations of VTR have long been identified as informative in the classification of Cryptobranchoidea (*Estes, 1981*; *Zhao and Hu, 1984*; *Fei et al., 2006*) and were recently argued to be ecologically informative (*Jia et al., 2021b*). To test if non-shape variations of the palate are reliable indicators in reconstructing paleoecology of early salamanders, we chose the following five continuous and two categorical covariates: length ratio of the vomer in the palate; length ratio of the parasphenoid in the palate; vomerine tooth number on a single vomer; width ratio of VTR in the vomer; width ratio of the outer and inner branch of VTR; and both the position (relative to the vomerine plate: anterior,

middle, mid-posterior or posterior) and arrangement (oblique, transverse, curved) of VTR. Measurements of these continuous and categorical features were collected by using VG Studio Max and the Fiji platform (version 1.8.0_172; *Schneider et al., 2012*), with missing values for *Aviturus*, *Kokartus*, and *Pangerpeton* reconstructed by using the function 'phylopars' in 'Rphylopars' R package (version 0.3.2; *Goolsby et al., 2021*). Values and states for the five continuous covariates were visualized by boxplot in 'ggplot2' and were compared among different ecological groups by using Mann-Whitney U-test (*Figure 3*). Association of the two categorical covariates with ecology were investigated by contingency table and Cochran-Mantel-Haenszel test (*Figure 3—figure supplement 1* and *Supplementary file 1I*).

## Ancestral state reconstruction and evolutionary rate

The ancestral shapes of the palate for several internal nodes including the respective last common ancestors of Hynobiidae, Panhynobia, Pancryptobrancha, Cryptobranchoidea, Urodela, and Caudata were estimated by using maximum likelihood method in the 'anc.recon' function in the 'Rphylopars' R package (version 0.3.2; *Goolsby et al., 2021*). The ancestral value of the five continuous covariates for the internal nodes was estimated by using 'phylopars' in the same package (*Supplementary file 1J*). On the other hand, the ancestral states of the two categorical covariates and life history strategy were estimated using the function 'ace' in 'ape', with their likelihood for each of the internal node depicted as a pie chart (*Figure 3*). Wire plots for the ancestral shape of the internal nodes with reference to the mean shape were created by using the functions 'mshape' and 'plotRefToTarget' in 'geomorph'.

Evolutionary rates for each of the 24 landmarks collected on the palate, the vomer, the parasphenoid, and the five continuous covariates from the palate were calculated by using the function 'compare.multi.evol.rates' in 'geomorph'. Then evolutionary rates of the palate, the vomer and the parasphenoid among species across groups in ecology, life history strategy and taxonomic affiliation were calculated and compared by using the function 'compare.evol.rates' in 'geomorph' (*Supplementary file 1H*).

Results derived from this study were exported from R and were illustrated and assembled in Adobe Photoshop CC. Source codes for R used in this study is available at GitHub (https://github.com/SalamanderGeomorph/Salamander_Palate, copy archived at swh:1:rev:8cbdd82025b0bf987b-b6211239a2e7dc56c615d5, *Salaman, 2022*).

## Acknowledgements

We would like to thank C-F Zhou (Shandong University of Science and Technology) for help with field work and stratigraphic data collection; J Anderson (University of Calgary) and J Gardner (Royal Tyrrell Museum) for insightful comments and helpful discussions; J-P Jiang (Chengdu Institute of Biology, [CIB]), A Resetar (Field Museum of Natural History, [FMNH]), C-S Chen (Zhejiang Museum of Natural History), R-C Xiong (Liupanshui Normal University), G Wei (Zunyi Medical University) and S Wang (Capital Normal University) for access to comparative specimens of living salamanders under their curation. We are grateful to Z-X Luo, J Lemberg, A I Neander (all University of Chicago) and M-H Zhang (CIB) for their assistance in CT scanning specimens. We greatly appreciate *eLife* to generously waive the publication fee for us, the Senior Editor George Perry, Reviewing Editor and Reviewer Min Zhu and two other Reviewers Pavel Skutschas and David Marjanović for their valuable suggestions. JJ and GL are grateful to J-H JiaLi for her encouragement during the pandemic. JJ is supported by National Natural Science Foundation of China (41702002) and State Key Laboratory of Palaeobiology and Stratigraphy (Nanjing Institute of Geology and Palaeontology, CAS) (193111). K-QG is supported by National Natural Science Foundation of China (41872008).

## Additional information

### Funding

| Funder | Grant reference number | Author |
|---|---|---|
| National Natural Science Foundation of China | 41702002 | Jia Jia |
| National Natural Science Foundation of China | 41872008 | Ke-Qin Gao |
| State Key Laboratory of Palaeobiology and Stratigraphy | 193111 | Jia Jia |

The funders had no role in study design, data collection and interpretation, or the decision to submit the work for publication.

### Author contributions

Jia Jia, Conceptualization, Data curation, Formal analysis, Funding acquisition, Investigation, Methodology, Project administration, Resources, Software, Supervision, Validation, Visualization, Writing - original draft, Writing – review and editing; Guangzhao Li, Formal analysis, Methodology, Software, Validation; Ke-Qin Gao, Data curation, Funding acquisition, Resources, Writing – review and editing

### Author ORCIDs

Jia Jia http://orcid.org/0000-0002-8243-0156
Guangzhao Li http://orcid.org/0000-0002-5007-8338

### Decision letter and Author response

Decision letter https://doi.org/10.7554/eLife.76864.sa1
Author response https://doi.org/10.7554/eLife.76864.sa2

## Additional files

### Supplementary files

• Supplementary file 1. (A) Scores of the principal components generated from the standard principal component analysis (PCA) on the 24-landmark-dataset across 70 specimens, and the phylomorphospace analysis (PA), the phylogenetically aligned component analysis (PaCA), and the phylogenetic principal component analysis (Phylo-PCA) across 34 species. Abbreviations: Cumu. RV, Cumulative RV; Cumu. P., Cumulative Proportion; Eigenva., Eigenvalue; Prop. Co., Proportion of Covariance; Prop. V., Proportion of Variance; RV by Co., RV by Component; Sing. V., Singular Value. (B) Pairwise comparison and corresponding *p*-values of morphological disparity of the palate calculated as Procrustes variances for 34 species grouped by ecological preference, life history strategy and taxonomic affiliation. *, *p*-value ≤ 0.05. Abbreviations: cryptobran., cryptobranchoids. (C) Pairwise comparison and corresponding p-values of single landmark point of the palate calculated as Procrustes variances (×E-4) for 34 species grouped by ecological preference. *, p≤0.05; **, p≤0.01; ***, p≤0.001. (D) Pairwise comparison and corresponding *p*-values of single landmark point of the palate calculated as Procrustes variances (×E-5) for 34 species grouped by life history strategy. *, p≤0.05. (E) Pairwise comparison and corresponding p-values of single landmark point of the palate calculated as Procrustes variances (×E-5) for 34 species grouped by taxonomic affiliation. *, p≤0.05; **, p≤0.01. (F) Absolute morphological disparity of the palate calculated as Procrustes variances for 34 species grouped by ecology, life history strategy and taxonomic affiliations. (G) Absolute morphological disparity calculated as Procrustes variances and evolutionary rates for each landmark point of the palate across 34 species grouped by ecology, life history and taxonomic affiliations. (H) Comparison of evolutionary rates of the palate and centroid size for 34 species grouped by ecology, life history and taxonomic affiliations. *, p≤0.05; **, p≤0.01; ***, p≤0.001. (I) Contingency table showing the association between ecological preference and life history strategy and two discrete characters of vomerine tooth row (position and arrangement pattern) across 34 species represented by 70 specimens. (J) non-shape covariates of the palate of 35 species and the ancestral states for internal nodes reconstructed by the "Rphylopars" R package. (K) Definition of the 24 landmarks for the palate (vomer and parasphenoid) used in this study.

• Transparent reporting form

## Data availability

All data needed to evaluate the conclusions are included in the manuscript and the Supplementary file 1. Details of specimens, CT parameters and raw landmark coordinates and centroid sizes are available in three CSV files in the online Dryad repository (https://doi.org/10.5061/dryad.c59zw3r8x). Source codes for R and SAS used in this study are available at GitHub (https://github.com/SalamanderGeomorph/Salamander_Palate, copy archived at swh:1:rev:8cbdd82025b0bf987bb6211239a2e7dc56c615d5).

The following dataset was generated:

| Author(s) | Year | Dataset title | Dataset URL | Database and Identifier |
|---|---|---|---|---|
| Jia J, Li G, Gao K | 2022 | Specimen list and landmark coordinates for the palate of early salamanders | https:/doi.org/10.5061/dryad.c59zw3r8x | Dryad Digital Repository, 10.5061/dryad.c59zw3r8x |

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
