## [Editor Report]

This paper is a valuable contribution to evolutionary ecomorphology in extant and extinct tetrapods, and of interest to vertebrate paleontologists and other evolutionary biologists interested in the early evolution of amphibians. Using geometric morphometric analysis, the authors demonstrate that both the shape of the palate and several non-shape variables (particularly associated with vomerine teeth) are ecologically informative in early stem- and basal crown-group salamanders. The study also reveals that metamorphosis is significant in the expansion of ecomorphospace of the palate in early salamanders.

---

## [Decision Letter]

**Decision letter after peer review:**

Thank you for submitting your article "Palatal morphology predicts the paleobiology of early salamanders" for consideration by *eLife*. Your article has been reviewed by 3 peer reviewers, including Min Zhu as the Reviewing Editor and Reviewer #1, and the evaluation has been overseen by a Reviewing Editor and George Perry as the Senior Editor. The following individuals involved in the review of your submission have agreed to reveal their identity: Pavel Skutschas (Reviewer #2); David Marjanovic (Reviewer #3).

Essential revisions:

Using geometric morphometric analysis, you demonstrate that both the shape of the palate and several non-shape variables (particularly associated with vomerine teeth) are ecologically informative in early stem- and basal crown-group salamanders. If the used phylogenetic tree is accurate, their conclusions are robust. Please discuss what pitfalls might have been encountered in constructing the tree. Please better summarize the innovative aspects of the work and pay attention to the minor points raised by the reviewers.

*Reviewer #1 (Recommendations for the authors):*

The manuscript in the present format has the obvious weakness in the summarization of innovative points and needs revisions and more stylistic works.

Title: This looks like an overstatement of the present research.

Abstract: More words on the background, and clear summarization of innovative points. The logic connection between the present points is loose.

Line 21: "phylotypic designs" is not suitable for evolution.

Line 29: The sentence needs re-written. 'living … representatives from the Triassic'?

Line 47: 'evolutionary history of paleoecology'?

Lines 50-51: "crown + stem" replaced by "total group".

Line 52: "as sister-group taxa" is redundant here.

Lines 65-66: What is the logic here? How about the record of stem urodeles? Early salamanders (Line 42) should include stem urodeles.

Line 82: Is Triassurus a stem urodele? To be mentioned here.

Line 103: 'basal' replaced by 'fossil' to parallel with 'living'.

Line 113: add 'a' before 'stepwise pattern'.

Lines 114-118: Redundant with the contents in the concluding paragraph of the main text. To be deleted.

Line 133: 95.15% among 70 specimens? To be clarified here.

Lines 240 and other places. What is the difference between 'variables' and 'covariates'?

Line 241: 'most of the five' represents how many? Delete 'most of'?

Line 243: branches.

Lines 298-299. Meaningless. What do you mean by 'unhelpful'?

Line 303: To be clarified.

Line 309: Replaced by 'paleoecological turnover.

Line 314: Replaced by 'resulting'.

Line 320: 'other' to be deleted?

Line 339: 'fossil' is redundant here.

Line 360: water current?

Line 428: source data for rate?

Lines 435-438: The concluding sentence should be re-formatted.

Lines 480, 484: check words 'partes', 'pas'.

*Reviewer #3 (Recommendations for the authors):*

This manuscript is a valuable contribution to evolutionary ecomorphology in extant and extinct tetrapods. I recommend publication in *eLife* after appropriate revision.

The conversion of the manuscript to PDF format has caused a few problems. The text has suffered from encoding issues: some colons and probably all dashes have been replaced by squares, and seemingly random parts of the text have been replaced by randomly selected all-caps letters superimposed with squares, or by squares and a lot of white space. As a consequence, there are parts of the manuscript I cannot evaluate because, in extreme cases, entire lines are missing. Please fix this problem before the next round of review. The lines that this concerns are 37, 69, 75, 100, 123, 143, 153, 157, 187, 193, 202, 211, 216, 221, 257, 276, 474, 503, 514-516 (these three lines are almost completely obliterated), 519, 533, 541, 556, 573, 574, 581, 583, 584, 590, 597, 598, 635, 636, 639, 643, 651, 659, 661, 676, 677, 680, 686, 688, 690, 693, 696, 699, 726, 728, 764, every page range in the references, 825, 826, 904, 905, 932, 967, 970, 980, 973, and the name of every figure supplement.

In a few places the writing is difficult to parse, slowing readers down unnecessarily.

The lack of sirenids and salamandroids elegantly avoids the problem of the phylogeny of early salamanders (see below), but it means that crown-group salamanders are represented only by cryptobranchoids and maybe Beiyanerpeton. This greatly restricts this study's ability to reconstruct the first crown-group salamander. Given that sirenids and salamandroids are sister-groups and that all known sirenids (Cretaceous to extant) are only partially metamorphosed (much like Andrias), it is possible that adding them (and a few unquestioned salamandroids) to the datasets would modify the conclusions. This should either be tested – which would require repeating all your phylogeny-informed analyses, ideally twice to account for different phylogenetic hypotheses – or the conclusions about the first crown-group salamander should be strongly deemphasized in the text.

Following earlier analyses by the first and the last author, the phylogenetic positions of all mentioned Mesozoic salamanders are simply stated as facts. I'm surprised the work of Rong et al. (2020) is nowhere cited. It showed that the existing datasets for phylogenetic analysis of early salamanders are riddled with too many inaccuracies and redundant characters to be reliable. While Rong et al. (2020) did not undertake the necessary complete review of these datasets, which means their conclusions are not wholly reliable either, they did show that modest improvements result in cladograms that show Chunerpeton (redescribed in that paper), Beiyanerpeton, Qinglongtriton and possibly all other Mesozoic Chinese salamanders outside the salamander crown group. The reconstruction not only of the first crown-group salamander, but also of the first crown-group cryptobranchoid is affected.

Rong et al. (2020) further pointed out the nomenclatural fact that "herpeton" is grammatically neuter and that therefore the International Code of Zoological Nomenclature automatically corrects a number of species names from "-is" (masculine or feminine) to "-e" (neuter). The authors and dates of the names are not affected by this.

You follow common usage among paleontologists since the early 1990s in calling the crown group of salamanders Urodela and the total group Caudata. Apparently without talking to anyone, Wake (2020) has defined the name Caudata as applying to the crown group of salamanders in a way that is valid under the International Code of Phylogenetic Nomenclature (Cantino and de Queiroz, 2020). Wake (2020) explicitly left the name Urodela undefined. It might be best if you mention this situation in a few words in the manuscript. In the longer run, beyond this manuscript, it would probably be best to ask the Committee on Phylogenetic Nomenclature for an emendation of the definition of Caudata. I'm a member of the Committee and would happily coauthor a paper for this purpose with you and other experts on salamanders.

Lines 30, 40: I don't think Chinlestegophis and Rileymillerus should be accepted as undoubted caecilians. But as it happens, an undoubted Late Triassic caecilian was recently announced in a published conference abstract by Kligman et al. (2021), so the oldest known caecilians are Late Triassic in age either way.

35: These two papers are neither the most recent nor in any other sense the most important ones on this subject. I recommend citing Pardo et al. (2017b) and Daza et al. (2020: Figure 4D, E, S13, S14) instead.

36: "2017b" is an error for "2017a".

37: The review paper by myself and Laurin (2013) is not up to date, being written long before completion of the large phylogenetic analysis by myself and Laurin (2019), let alone its update by Daza et al. (2020: Figure 4F, S15), not to mention the analysis of ontogeny by Laurin et al. (2022). Importantly, it is clear that Lissamphibia is not derived from aquatic lepospondyls.

40-42: This is plainly not true. Chinlestegophis obviously lived in burrows, but apart from its elongate body shape it shows no adaptations to burrowing. Even the Early Jurassic Eocaecilia and, as far as its fragmentary remains allow us to tell, the Early Cretaceous Rubricacaecilia lack some of the crown group's adaptations to burrowing, e.g. they retain limbs, larger orbits and more sutures in the skull.

49: Of any two sister groups, each is more primitive than the other in some respects but not in others. It makes little sense to call Cryptobranchoidea "the most primitive clade of the crown group salamanders". I suggest "the sister group to all other crown salamanders".

57, 625-626: For the life history of Aviturus, see Skutschas et al. (2018).

58-59: It is not excluded that Regalerpeton is a stem-group salamander (Rong et al. 2020: Figure 5).

71, 73: Two more good opportunities to cite Rong et al. (2020).

89, 173-179, 446-450, 615: but see Rong et al. (2020) for reasons for skepticism.

110, 198, 268, 271, 276, 277, 279, 280, 282, 284, 286, 289, 290, 334, 337, 428: Insert "last" before "common"!

194: Replace "albert" by "albeit" (or "although" or just "though").

202, 602, 635: Uppercase for Procrustes (the name of a mythological person).

293-294: All adult anurans and caecilians are metamorphosed, but some of both are fully aquatic (e.g. Pipidae, Typhlonectidae). I would therefore rearrange the sentence to: "Among modern amphibians, the adults of anurans and caecilians are metamorphosed, and most of them are terrestrial."

299: …or whatever will remain of Lepospondyli.

308: I would write "metamorphic taxa" or "metamorphosed individuals".

339-340: Aviturus, which dates from the very end of the Paleocene, is not the earliest known pancryptobranchan. The earliest entirely undoubted one is "Cryptobranchus" saskatchewanensis, which is a few million years older (late middle Paleocene). There is evidence that the much older (mid-Cretaceous) Eoscapherpeton is a stem-pancryptobranchan; see Marjanović (2021: supplementary material pp. 13-16) for references and a brief review.

480: The plural of pars palatina is partes palatinae – as in most languages of Europe (but unlike English), number and gender (and case) are marked on adjectives as well as nouns in Latin.

484: Replace "pas" by "pars".

791: Replace "batrachian" by "batrachians".

857-859: Unlike English, German does not have separate capitalization rules for headlines – but it always gives a capital Letter to every Noun, while almost nothing else is ever capitalized, not even Adjectives derived from proper Names. Therefore, please correct the Title to: "Generelle Morphologie der Organismen: Allgemeine Grundzüge der organischen Formen-Wissenschaft, mechanisch begründet durch die von Charles Darwin reformirte Descendenz-Theorie".

Finally, the supplementary file claims to be a PDF file ("This PDF file contains supplementary file 1A to 1K."), but it is a DOCX file. In the interest of wider accessibility, I recommend you convert it to PDF before resubmitting it.

I hope these comments are helpful. I apologize again for the delay, and I'm looking forward to the next version of your manuscript!

References not cited in the manuscript:

Cantino PD, de Queiroz K. 2020. International Code of Phylogenetic Nomenclature (PhyloCode). Version 6. CRC/Taylor & Francis/Informa, Boca Raton/London/New York. ISBN: 978-1-138-33282-9 (paperback), 978-1-138-33286-7 (hardback), 9780429446320 (e-book). DOI: https://doi.org/10.1201/9780429446320 Openly accessible at http://phylonames.org/code/

Daza JD, Stanley EL, Bolet A, Bauer AM, Arias JS, Čerňanský A, Bevitt JJ, Wagner P, Evans SE. 2020. Enigmatic amphibians in mid-Cretaceous amber were chameleon-like ballistic feeders. Science 370:687-691. DOI: https://doi.org/10.1126/science.abb6005

Kligman B, Stocker M, March A, Nesbitt S, Parker W. 2021. New Late Triassic stem-caecilian from southwestern North America strengthens evidence for lissamphibian monophyly, and illuminates the anatomical, functional and geographic origins of living caecilians [abstract]. Society of Vertebrate Paleontology (ed.): Virtual meeting conference program, 81st annual meeting, p. 160. The entire abstract volume can be downloaded here: https://vertpaleo.org/svp_2021_virtualbook_final/

Laurin M, Lapauze O, Marjanović D. 2022. What do ossification sequences tell us about the origin of extant amphibians? Peer Community Journal 2:e12. DOI: https://doi.org/10.24072/pcjournal.89

Marjanović D. 2021. The making of calibration sausage exemplified by recalibrating the transcriptomic timetree of jawed vertebrates. Frontiers in Genetics 12:521693. DOI: 10.3389/fgene.2021.521693

Marjanović D, Laurin M. 2019. Phylogeny of Paleozoic limbed vertebrates reassessed through revision and expansion of the largest published relevant data matrix. PeerJ 6:e5565. DOI: https://doi.org/10.7717/peerj.5565

Rong Y-F, Vasilyan D, Dong L-P, Wang Y. 2020 (printed 2021). Revision of Chunerpeton tianyiense (Lissamphibia, Caudata): Is it a cryptobranchid salamander? Palaeoworld 30:708-723. DOI: https://doi.org/10.1016/j.palwor.2020.12.001

Skutschas PP, Kolchanov VV, Bulanov VV, Sennikov AG, Boitsova EA, Gulbev VK, Syromyatnikova EV. 2018 (printed 2020). Reconstruction of the life history traits in the giant salamander Aviturus exsecratus (Caudata, Cryptobranchidae) from the Paleocene of Mongolia using zygapophyseal skeletochronology. Historical Biology 32:645-648. DOI: https://doi.org/10.1080/08912963.2018.1523157

Wake DB. 2020. Caudata J. A. Scopoli 1777 [D. Wake], converted clade name. [Brackets in the original.] de Queiroz K, Cantino PD, Bauthier JA (eds): Phylonyms. A Companion to the PhyloCode (CRC/Taylor & Francis/Informa, Boca Raton/London/New York), pp. 785-787. ISBN: 978-1-138-33293-5 (hardback), 9780429446276 (e-book). DOI of the entire book: https://doi.org/10.1201/9780429446276

[Editors' note: further revisions were suggested prior to acceptance, as described below.]

Thank you for resubmitting your work entitled "Palatal morphology predicts the paleobiology of early salamanders" for further consideration by *eLife*. Your revised article has been evaluated by George Perry (Senior Editor) and a Reviewing Editor.

The manuscript has been improved but there are some remaining issues that need to be addressed, as outlined below:

*Reviewer #1:*

They have addressed my concerns in my previous review and revised their manuscript accordingly. I have no additional recommendations for the authors.

*Reviewer #3:*

The issues with image quality in the merged PDF have disappeared. Please correct the species names in figures 1, 3, 19, 23-26, 30-32, 36 and 38 if I've counted correctly, and replace "semiaqaic" with "semiaquatic" in Figure 3.

Sirenidae and the first crown-group salamander

Your anatomical reasons for the omission of sirenids are convincing, and I had indeed confused sirenids and amphiumids in my statement about partial metamorphosis – thank you for pointing this out! I am, however, quite surprised that you present the phylogenetic position of Sirenidae as a mystery. Sirenidae and Salamandroidea have been found as sister groups in every large-scale study of molecular data, no matter which data or which method exactly: 7,189 transcripts of nuclear genes and Bayesian inference (Irisarri et al. 2017; 99% posterior probability), 5 mitochondrial and 10 nuclear genes and maximum likelihood (Vijayakumar et al. 2019: supplementary file Amphibia_New_India_SHL_Dryad.tre; 52% bootstrap support – note this is the latest and largest version of R. A. Pyron's series of matrices, and all versions, including Jetz and Pyron [2018] which you cite, found the same topology), or 120 nuclear protein-coding genes, separately and together, and maximum likelihood (Himes et al. 2020: 100% bootstrap and ASTRAL posterior values). Given such extraordinary agreement, the fact that the morphological evidence is less clear can be blamed on the morphological evidence, or on our lack of knowledge of it (I greatly appreciate your recent work in describing the osteology of extant salamanders!), but it would be very difficult to try to argue the molecular data away. Therefore, I think you should mention in the text that you cannot include Sirenidae in your dataset for the reasons you presented in your response and that this is a potential weak point in inferences about the last common ancestor of all crown-group salamanders.

You point out that the analyses of Rong et al. (2020, printed 2021) failed to find Hynobiidae or Cryptobranchoidea. Unfortunately, this is not incompatible with their matrix being an improvement over its sources. It is not uncommon (e.g. Marjanović and Laurin, 2019, and references therein) for improvements to matrices for phylogenetic analysis to decrease resolution, and it is not uncommon either for matrices to support the right things for the wrong reasons. Matrices that support clades for the wrong reasons are unreliable in the sense that they cannot be trusted to place added taxa accurately. As I stated, the version by Rong et al. (2020) cannot be relied upon as strong evidence for the phylogenetic positions of Chunerpeton, Beiyanerpeton or any other taxon – but that does not make the earlier versions of that matrix any better. They contain, after all, such phenomena as redundant and even duplicate characters: for example, the haploid and the diploid number of chromosomes are both included as separate characters in the previous versions. Duplicating a character inevitably distorts, if not the topology, then at the very least the support values for that topology (references in Marjanović and Laurin, 2019: 15-16). Rong et al. (2020) made a certain effort to reduce this problem.

They also ordered certain characters, unlike their sources. Ordering is widely believed to be a philosophical question, but there is strong evidence from simulations as well as from empirical studies that potentially clinal or meristic characters must be ordered to avoid inaccurate results as well as both false negatives and false positives in resolution (references and brief discussion in Marjanović and Laurin, 2019: 16).

Phylogenetics with morphological data

Parsimony analysis of morphological data should not be treated as a black box.

Daza et al. (2020) did all their phylogenetic work on previously published matrices; they added or updated the scores of Albanerpetidae, but did not make any other changes to those matrices. Therefore, practically all the problems with their results are the fault of those matrices. The clearly wrong placement of Chelotriton in their Figure 4F and S15 stem from the fact that I had added Chelotriton to the source matrix in question, that of Marjanović and Laurin (2019), which is simply not equipped to handle crown-group salamanders. (That matrix is an attempt to improve the one by Ruta and Coates, 2007. In order to limit the amount of time and effort, we did not add any characters to that matrix. The matrix of Ruta and Coates, 2007, contained only two salamanders – Karaurus and Valdotriton – and therefore lacked any characters specific to salamander phylogeny.) I did this in order to test a point on ontogeny and phylogeny: would the extreme metamorphosis of Chelotriton, which makes it look more like a temnospondyl than lissamphibians usually do, pull some or all lissamphibians into Temnospondyli? That did not happen, but Chelotriton was pulled out of the salamander clade. The latter result is obviously wrong, and obviously due to the lack of salamander-specific characters in the matrix; there is no reason to think that it indicates a more general problem. The former result, on the other hand, is actually strengthened that way.

In another analysis, Daza et al. (2020: Figure 4E, S14) found Karauridae outside Batrachia. I haven't looked into its precise causes, but it is simply shared with the source matrix of that analysis, the matrix of Pardo et al. (2017a).

Both of these analyses are, however, relevant to the ecology of the first lissamphibian by finding Albanerpetidae as the sister group of Lissamphibia, Albanerpetidae being terrestrial and fully metamorphic or of course direct-developing (early juveniles of any sort are not known). This is a data point for phylogenetic bracketing.

In the context of the specimens referred to Chunerpeton by Rong et al. (2020), you write: "only those morphological features in adults are trustworthy for taxonomic and phylogenetic interpretations." In phylogenetics, however, morphological features are not automatically comparable just because they are found in sexually mature adults, as you seem to have assumed in your published phylogenetic work: for much of the skeleton, for instance, neotenic adults are much more easily comparable to larvae than to adults of metamorphic taxa, so that the presence or absence of many character states of metamorphic adults must be scored as unknown/inapplicable in neotenic taxa. This has a strong effect on the topologies found by phylogenetic analyses (Wiens et al., 2005; Marjanović and Laurin, 2019: 21-22, and references therein).

Nomenclature and taxonomy

I greatly appreciate your comments on the validity of the referral of the new specimens by Rong et al. (2020) to Chunerpeton, and I'm looking forward to your further publications on this matter!

Likewise, I appreciate your comments on the definitions of the names Caudata and Urodela and will try to send you a draft manuscript as soon as possible. For the purposes of the present manuscript, I withdraw all my objections.

Adaptations of Chinlestegophis

Thank you for reminding me of the features listed as adaptations to burrowing by Pardo et al. (2017a). I agree that the fairly small orbits may count as such, though they could also hint at life at the bottom of muddy bodies of water – a strictly aquatic lifestyle, apart from sheltering in burrows with 100% humidity, is assured by the lateral-line grooves. The position of the jaw articulation, however, is shared with all other brachiopods, where it is an adaptation to a particular style of suction feeding; it is in fact more extreme in Batrachosuchus, depicted in Pardo et al. (2017a: Figure 3). By "consolidation of the skull", Pardo et al. (2017a: supp. inf. part F) mean the frankly irrelevant fusion of lacrimal and maxilla – besides, it is not in fact clear whether a lacrimal is even present -, the supposed fusion of the pterygoid and the quadrate for which there is very little evidence presented in the paper, and fusion of the exoccipitals and the basioccipital which is universal in dissorophoid temnospondyls, none of which were burrowing or had any sort of digging lifestyle – and note that it is not clear if a basioccipital was present in the first place; it remained cartilaginous and very small in other brachiopods. In a borrower, one would expect, as in extant caecilians, a well-ossified braincase that could function as a strut to enable the skull to resist rostrocaudal compression, dorsoventral bending, or twisting; yet, most of the braincase is not ossified at all in Chinlestegophis (Pardo et al. 2017: supp. inf. part B). This is particularly striking in comparison to the burrowing "lepospondyls" described by Pardo and various coauthors in the two years prior.

Interestingly, the stem-caecilian presented by Kligman et al. (2021) seems not to be adapted to burrowing at all, but it is clearly much closer to the crown group than Chinlestegophis.

Details

Lines 38-41: I had not quite appreciated that here you cite references both for the lifestyles and the phylogenetic positions of the listed taxa, and only suggested references for the phylogenetic positions of various "lepospondyls". For the terrestrial lifestyle of at least one amphibamid, I recommend Laurin et al. (2004) and references therein. For the various lifestyles of "lepospondyls", I recommend Jansen & Marjanović (2021) and references therein. While some stereospondyls have occasionally been considered semiaquatic for vaguely articulated reasons, almost all were certainly fully aquatic as shown by the lateral-line grooves on their skulls and further supported by their very slow peri- and endochondral ossification; I recommend the brief but clear statement in Schoch & Milner (2014: 123) and references therein. Notably, Chinlestegophis has lateral-line grooves (Pardo et al., 2017a), showing that it was not semiaquatic; "reduced" grooves as identified by Pardo et al. (2017a) mean that most of the lateral-line organ was situated in the skin and did not contact the bone, not that the organ was "reduced" – something that does not occur, because the lateral-line organ dries up and dies from serious exposure to air as inevitably caused by a semiaquatic lifestyle. – While Jansen & Marjanović (2021) is published on a preprint server with the usual disclaimer, it is in fact an accepted manuscript published with permission from the journal (Comptes Rendus Palevol) that accepted it after peer review. The journal is currently changing publishers, a process that started before the pandemic and is still not complete; we've been waiting for the page proofs since March 2021.

80-81: Other than its probable membership in Karauridae, is there evidence on the lifestyle of Marmorerpeton?

85, 106: If you use Caudata as the name for the total group, Triassurus is a stem caudate, not a stem urodele; nothing is a stem urodele, because if something is on the stem from which Urodela comes, it is outside Urodela by definition. At least for the first few decades of this terminology, a clade consisted of a crown group and a stem-group; to be a stem urodele, something has to be a urodele.

106: Likewise, if you use Urodela as the name for the crown group, "crown urodeles" is redundant; "urodeles" would be enough.

623-624: I would rather write: "As implied by Jia et al. (2021a), we apply the name Hynobiidae to the crown group of Panhynobia".

912: "lepospondyl", not "lepospondyls".

928: "Kuro-o", a Japanese name with three syllables; "oo" could be misunderstood as a long vowel – Japanese distinguishes long from short vowels. Sometimes apostrophes are used to disambiguate in transcriptions, sometimes hyphens are used instead.

976: "Syromyatnikova".

977: italics for the genus & species name (present in the original, I've checked)

Suggestions on style and language

Lines 15-18: "but the small number of reliable ecological indicators established so far hinders investigations into the paleobiology of early salamanders. Here we statistically demonstrate, by using time-calibrated phylogenetic trees and geometric morphometric analysis on 71 specimens in 36 species, that both the shape"…

21: I'm not sure what you mean by "strictly". Perhaps "analyzed in detail" would be clearer?

25-27: That would mean the disparities, not the salamanders, have achieved the ecological preferences. Would the following be an improvement? "Salamanders began to diversify ecologically before the Middle Jurassic and achieved all their present modes of life in the Early Cretaceous."

References cited above but not in the manuscript

Hime PM, Lemmon AR, Moriarty Lemmon EC, Prendini E, Brown JM, Thomson RC, Kratovil JD, Noonan BP, Pyron RA, Peloso PLV, Kortyna ML, Keogh, JS, Donnellan SC, Lockridge Mueller R, Raxworthy CR, Kunte K, Ron SR, Das S, Gaitonde N, Green DM, Labisko J, Che J, Weisrock DW. 2020. Phylogenomics reveals ancient gene tree discordance in the amphibian tree of life. Systematic Biology 70:49-66. DOI: 10.1093/sysbio/syaa034

Laurin M, Girondot M, Loth M-M. 2004. The evolution of long bone microstructure and lifestyle in lissamphibians. Paleobiology 30:589-613. DOI: 10.1666/0094-8373(2004)030<0589:TEOLBM>2.0.CO;2

Irisarri I, Baurain D, Brinkmann H, Delsuc F, Sire J-Y, Kupfer A, Petersen J, Jarek M, Meyer A, Vences M, Philippe H. 2017. Phylotranscriptomic consolidation of the jawed vertebrate timetree. Nature Ecology & Evolution 1:1370-1378. DOI: 10.1038/s41559-017-0240-5

Jansen M, Marjanović D. 2021. The scratch-digging lifestyle of the Permian "microsaur" Batropetes as a model for the exaptative origin of jumping locomotion in frogs. bioRχiv 460658. DOI: 10.1101/2021.09.27.460658

Schoch RR, Milner AR. 2014. Temnospondyli I. Part 3A2 of Sues H-D (ed.): Handbook of Paleoherpetology. München: Dr. Friedrich Pfeil.

Vijayakumar SP, Pyron RA, Dinesh KP, Torsekar VR, Srikanthan AN, Swamy P, Stanley EL, Blackburn DC, Shanker K. 2019. A new ancient lineage of frog (Anura: Nyctibatrachidae: Astrobatrachinae subfam. nov.) endemic to the Western Ghats of Peninsular India. PeerJ 7:e6457. DOI: 10.7717/peerj.6457

---

## [Author Response]

Essential revisions:Using geometric morphometric analysis, you demonstrate that both the shape of the palate and several non-shape variables (particularly associated with vomerine teeth) are ecologically informative in early stem- and basal crown-group salamanders. If the used phylogenetic tree is accurate, their conclusions are robust. Please discuss what pitfalls might have been encountered in constructing the tree. Please better summarize the innovative aspects of the work and pay attention to the minor points raised by the reviewers.Reviewer #1 (Recommendations for the authors):The manuscript in the present format has the obvious weakness in the summarization of innovative points and needs revisions and more stylistic works.Title: This looks like an overstatement of the present research.

We pondered this comment and we firmly believe this title does justice to the aim, analyzes and conclusions of our manuscript. We believe that the top priority to understand the evolutionary paleobiology of early salamanders in this study is to establish osteological indicators for ecology in living taxa, which are the shape of the palate and seven covariates associated with the vomer and vomerine teeth, and demonstrate that these indicators are reliable through a set of strict statistical analyses. Then these ecological indicators are applied to fossil taxa to recover their paleoecological preference and reconstruct the ancestral configurations of these indicators in the hypothesized ancestors of different salamander clades.

Abstract: More words on the background, and clear summarization of innovative points. The logic connection between the present points is loose.

Please see the revised Abstract section in the resubmitted manuscript. We appreciate this advice.

Line 21: "phylotypic designs" is not suitable for evolution.

To avoid confusions, “phenotypic designs” is now replaced by “phenotypic configurations” in Line 21 and Line 378.

Line 29: The sentence needs re-written. 'living … representatives from the Triassic'?

The sentence “Salamanders, anurans and caecilians are highly distinctive from one another in their morphology in both living and geologically the oldest known representatives from the Triassic” is now re-written as “Salamanders, anurans and caecilians are highly distinctive from one another in their morphology in both living species and their respective oldest known relatives from the Triassic”.

Line 47: 'evolutionary history of paleoecology'?

“evolutionary history of paleoecology” changed to “evolutionary paleoecology”.

Lines 50-51: "crown + stem" replaced by "total group".

done.

Line 52: "as sister-group taxa" is redundant here.

“as sister-group taxa” is deleted.

Lines 65-66: What is the logic here? How about the record of stem urodeles? Early salamanders (Line 42) should include stem urodeles.

Right, early salamanders do include both stem and basal crown urodeles. We did originally include introductions on stem urodeles in the posterior half of this paragraph, starting from the sentence----“In contrast, other contemporaries (e.g., *Kokartus*, *Marmorerpeton*) from the Middle Jurassic Bathonian of UK, Russia and Kyrgyzstan are all neotenic and aquatic at their adult stage, and have been classified as stem urodeles by…”

We reworded the first sentence of this paragraph to better emphasize the importance of Cryptobranchoidea in understanding the early paleoecology of salamanders, and now it goes like this----“Cryptobranchoidea are critical in understanding the paleoecology of early salamanders because the earliest known cryptobranchoids from the Middle Jurassic Bathonian have higher disparities in both life history strategies and ecological preferences than stem urodeles, and represent the oldest known crown urodeles, including ‘Kirtlington salamander B’ from the UK,….”.

Line 82: Is Triassurus a stem urodele? To be mentioned here.

It is a stem urodele. “The only known pre-Jurassic salamander, *Triassurus*…” is now replaced by “The only known pre-Jurassic stem urodele, *Triassurus*…”.

Line 103: 'basal' replaced by 'fossil' to parallel with 'living'.

We replaced the first “basal” in “basal genera of cryptobranchoids” by “fossil”, and kept the second “basal” in “stem and basal crown urodeles” to parallel with “stem”.

Line 113: add 'a' before 'stepwise pattern'.

done.

Lines 114-118: Redundant with the contents in the concluding paragraph of the main text. To be deleted.

The two sentences at Lines 114-118 are now deleted.

Line 133: 95.15% among 70 specimens? To be clarified here.

Correct. We added “in 70 specimens” after “96.15% of the total shape variation”.

Lines 240 and other places. What is the difference between 'variables' and 'covariates'?

In statistics, both “variables” and “covariates” refer to characteristics of the participants in an experiment. Both can be used to refer independent variables, whereas covariates sometimes would have another layer of meaning when referring to the interrelationships among variables. We use “covariates” for the five continuous and two categorial non-shape variables and to differentiate them from the shape variables, which are 2D landmark points represented by x and y variables. We think “covariate” is accurate for non-shape variables because these seven covariates are mostly derived from the same subset of the palate (vomer + parasphenoid), the vomer/vomerine teeth, and any changes in the configuration of vomer/vomerine teeth can affect most of these covariates. So theses covariates are sort of “inter-related”. To be consistent, we replaced “variables” with “covariates” for non-shape variables in the manuscript.

Line 241: 'most of the five' represents how many? Delete 'most of'?

Only the first four out of the five continuous covariates listed within the parentheses are significantly impacted by allometry, not the fifth covariate (vomerine tooth number), and now we replaced “most” by “the first four”.

Line 243: branches.

done.

Lines 298-299. Meaningless. What do you mean by 'unhelpful'?

The earliest lissamphibians should have a unified lifestyle if they have a monophyletic origin. “unhelpful” here refers to the conflict between this hypothesized unified lifestyle in the earliest lissamphibians and the diverse lifestyles in modern lissamphibians. We now revised the sentence as the following: “The discrepancy in ecological preference between salamanders and anurans and caecilians is unhelpful in understanding the evolutionary paleoecology in the early lissamphibians given that they have a monophyletic origin from Temnospondyli.”

Line 303: To be clarified.

“…which to some extent are contributed by the lack of taphonomic analyses (Wang et al., 2019)” is now changed to “…which to some extent are contributed by the insufficient taphonomic analyses on fossil sites with salamander discoveries, such as the Daohugou fossil locality (Wang et al., 2019)”.

Line 309: Replaced by 'paleoecological turnover.

done.

Line 314: Replaced by 'resulting'.

replaced as suggested.

Line 320: 'other' to be deleted?

done.

Line 339: 'fossil' is redundant here.

“fossil” is removed.

Line 360: water current?

“a current of water” is replaced with “water current”

Line 428: source data for rate?

We added “(Supplementary file 1H)” after “…have the highest evolutionary rate in the palate” to show the source data for this statement.

Lines 435-438: The concluding sentence should be re-formatted.

We now have reformatted the concluding sentence as below: “Our results rigorously show that the shape of the palate and many non-shape covariates particularly associated with vomerine teeth are reliable ecological indicators for paleoecology of early salamanders, and we demonstrate that metamorphosis with the biphasic ecological preference (aquatic larvae + terrestrial adults) is not only the ancestral lifestyle in salamanders but also significant for the rise and diversification of modern amphibians.”

Lines 480, 484: check words 'partes', 'pas'.

To match our use of “paired vomers and a single median parasphenoid” in the same sentence, we chose to use the plural form of pars palatina for “premaxilla and maxilla”. The word “pars” is a singular noun in Latin (means part in English) and “palatina” is a singular nominative declining adjective in Latin (means palatine in English), and we agree with Dave (Reviewer #3) on that Latin adjectives should follow nouns in number, gender as well as case. The plural form for “pars palatina” therefore should be “partes palatinae” and thus we corrected our mistake. However, it is confusing to find both “partes palatina” and “partes palatinae” in published studies found through Google Scholar.

We also corrected our typo “pas” and replaced it with “pars” at Line 485.

Reviewer #3 (Recommendations for the authors):This manuscript is a valuable contribution to evolutionary ecomorphology in extant and extinct tetrapods. I recommend publication in eLife after appropriate revision.

We thank you for your encouragement and please find our revisions in the re-submitted manuscript and our responses to your questions below.

The conversion of the manuscript to PDF format has caused a few problems. The text has suffered from encoding issues: some colons and probably all dashes have been replaced by squares, and seemingly random parts of the text have been replaced by randomly selected all-caps letters superimposed with squares, or by squares and a lot of white space. As a consequence, there are parts of the manuscript I cannot evaluate because, in extreme cases, entire lines are missing. Please fix this problem before the next round of review. The lines that this concerns are 37, 69, 75, 100, 123, 143, 153, 157, 187, 193, 202, 211, 216, 221, 257, 276, 474, 503, 514-516 (these three lines are almost completely obliterated), 519, 533, 541, 556, 573, 574, 581, 583, 584, 590, 597, 598, 635, 636, 639, 643, 651, 659, 661, 676, 677, 680, 686, 688, 690, 693, 696, 699, 726, 728, 764, every page range in the references, 825, 826, 904, 905, 932, 967, 970, 980, 973, and the name of every figure supplement.In a few places the writing is difficult to parse, slowing readers down unnecessarily.

We apologize for any inconvenience caused by the flawed PDF and thank you for pointing out the places with words loss and format chaos. We believe the merged PDF is not the one we originally approved (48 MB in size) and instead is likely a size-compressed copy (~1.46 MB in size) uploaded to the Biorxiv Preprint platform (version 1). However, the Word file of our manuscript that we originally submitted works well though, and we also immediately uploaded a new PDF (as version 2; https://www.biorxiv.org/content/10.1101/2022.01.17.476642v2) to Biorxiv on January 22, 2022 when we noticed the original PDF was not working well.

The lack of sirenids and salamandroids elegantly avoids the problem of the phylogeny of early salamanders (see below), but it means that crown-group salamanders are represented only by cryptobranchoids and maybe Beiyanerpeton. This greatly restricts this study's ability to reconstruct the first crown-group salamander. Given that sirenids and salamandroids are sister-groups and that all known sirenids (Cretaceous to extant) are only partially metamorphosed (much like Andrias), it is possible that adding them (and a few unquestioned salamandroids) to the datasets would modify the conclusions. This should either be tested – which would require repeating all your phylogeny-informed analyses, ideally twice to account for different phylogenetic hypotheses – or the conclusions about the first crown-group salamander should be strongly deemphasized in the text.

The reasons we refuse to add sirenids into our dataset are three folds. First, living sirenids (*Siren* + *Pseudobranchus*) have many autapomorphies (e.g., loss of pelvic girdle + hind limb, separate scapular and coronoid) at the level of Caudata and numerous morphological specializations (e.g., extremely narrow snout, no teeth in upper jaw and dentary, a toothed coronoid) including features specifically related to our manuscript, such as presence of a patch of teeth on vomer and palatine formed by multiple rows of tiny teeth (Reilly and Altig, 1996; and see many datasets on the MorphoSource platform). However, the only known fossil sirenid *Habrosaurus* found from the latest Cretaceous (late Maastrichtian [72.1-66 Ma]) to Paleocene has a more general configuration in the skull (e.g., elongate maxilla, rudimentary upper jaw, teeth present in upper jaw and dentary; single tooth row on each vomer; but a larger palatine with even denser teeth than living sirenids) when compared to living sirenids (Gardner, 2003: Figure 3I-L; Figure 9), indicating the specialization of sirenids is not formed from the very beginning of their evolution, just like the evolution of morphological specializations of caecilian for their fossorial lifestyle as you mentioned elsewhere. Unfortunately, specimens attributed to *Habrosaurus* are too poorly-preserved to allow a full restoration of the skull, and the palate remains incompletely known (parasphenoid is unknown Gardner, 2003), leaving it impossible to be added into our dataset. If Sirenidae does share a sister group relationship with Salamandroidea (geologically oldest representative *Beiyanerpeton* dates back to ~160 Ma), there would be a ~90 Ma fossil gap for Sirenidae that would greatly impact our understandings of their early evolution and the configurations of the palate.

Second, sirenids are probably the only herbivory salamanders (remaining salamanders are carnivory) and have recently been shown as having a unique way of intraoral food processing, which is probably different from all other lissamphibians (Schwartz et al., 2020, 2021): instead of swallowing food unreduced, sirenids have complex three-dimensional chewing behavior to extract energy from plant matters. In our opinion, sirenids might have biomechanical patterns different from all other salamanders during feeding to coordinate with the chewing behavior, and thus their specialized configurations in the palate (such as the enlarged palatine and the many palatine teeth [Gardner, 2003: Figure 3J]. Note that palatine is absent in almost all other salamander clades at the adult stage, except proteids) may receive different selection pressure on the palate of non-sirenid salamanders. The special feeding mode of sirenids and the absence of palatine and palatine teeth in most other salamander clades show that it would be inappropriate to include sirenids into our dataset.

Third, both the extreme specialization in morphology in sirenids and the rampant homoplasies shared with other neotenic taxa (see below) led to the fact that consensus about the phylogenetic position of sirenids has not been reached for over 130 years (Cope, 1889; Larson and Dimmick, 1993; Zhang and Wake, 2009). To date, sirenids are found either as the sister group taxon to Salamandroidea or nested within the latter at different positions.

We also do not agree with the statement “all known sirenids (Cretaceous to extant) are only partially metamorphosed (much like *Andrias*)”. Sirenids have long been recognized as obligate neotenic species as amphiumids and proteids and share many homoplasies stemming from their aquatic adaptations (e.g., elongate trunk, long and bushy external gills; e.g., Deban and Wake, 2000; Wake and Deban, 2000; Wake, 2009; Bonett and Blair, 2017) and have way more neotenic features than the partially metamorphosed cryptobranchids (*Andrias* + *Cryptobranchus*). Cryptobranchids are heavily metamorphosed and retain only few neotenic features, for example, external gills are lost in both *Andrias* and *Cryptobranchus*; *Cryptobranchus* retains gill slits whereas *Andrias* has the gill slits closed.

Clearly, with so many uncertainties and outstanding questions centering around sirenids, inclusion of sirenids into our dataset will only bring harm, heavily reduce the credibility of many results of our study and will for sure bring unreliable hypotheses including reconstructions of the palate shape.

For Salamandroidea, their fossil records are scarce in the Mesozoic and in our dataset we included the most primitive salamandroid *Beiyanerpeton* with the hope to add appropriate and relevant data for analyses, such as shape reconstructions of the palate for Urodela.

Some citations above are included in our manuscript and we listed those not included below:

Cope, E.D. 1889. The Batrachia of North America. Bulletin of the United States National Museum 34:1–515.

Gardner, J.D. 2003. Revision of *Habrosaurus* Gilmore (Caudata; Sirenidae) and relationships among sirenid salamanders. Palaeontology 46:1089–1122.

Larson, A., and W.W. Dimmick. 1993. Phylogenetic relationships of the salamander families: an analysis of congruence among morphological and molecular characters. Herpetological Monographs 7:77–93.

Reilly, S.M., and R. Altig. 1984. Cranial osteology in *Siren intermedia* (Caudata: Sirenidae): paedomorphic, metamorphic and novel patterns of heterochrony. Copeia 1996:29–41.

Schwartz, D., N. Konow, Y.T. Roba, and E. Heiss. 2020. A salamander that chews using complex, three-dimensional mandible movements. Journal of Experimental Biology 223:jeb220749.

Schwartz, D., M.T. Fedler, P. Lukas, A. Kupfer. 2021. Form and function of the feeding apparatus of sirenid salamanders (Caudata: Sirenidae): three-dimensional chewing and herbivory? Zoologischer Anzeiger 295:99–116.

Zhang, P., and D.B. Wake. 2009. Higher-level salamander relationships and divergence dates inferred from complete mitochondrial genome. Molecular Phylogenetics and Evolution 53:492–508.

Following earlier analyses by the first and the last author, the phylogenetic positions of all mentioned Mesozoic salamanders are simply stated as facts. I'm surprised the work of Rong et al. (2020) is nowhere cited. It showed that the existing datasets for phylogenetic analysis of early salamanders are riddled with too many inaccuracies and redundant characters to be reliable. While Rong et al. (2020) did not undertake the necessary complete review of these datasets, which means their conclusions are not wholly reliable either, they did show that modest improvements result in cladograms that show Chunerpeton (redescribed in that paper), Beiyanerpeton, Qinglongtriton and possibly all other Mesozoic Chinese salamanders outside the salamander crown group. The reconstruction not only of the first crown-group salamander, but also of the first crown-group cryptobranchoid is affected.

We believe the results of any cladistic analyses are hypotheses at best not facts, and we support our hypotheses with evidences at hand like other researchers support their own. And we clearly cited the source references for cladograms we used in this study.

Thank you for bringing up the work of Rong et al. (published online in 2020; printed in 2021). Throughout their article, we did not find any words supporting the above statement “existing datasets for phylogenetic analysis of early salamanders are riddled with too many inaccuracies and redundant characters to be reliable”, and the only relevant contents we found are two sentences below:

1. “For our first analysis, we modified fourteen codings (Appendix A, Section 3) for *Chunerpeton* in the matrix of Jia and Gao (2019), based on the new fossils we examined” (Rong et al., 2021: p.719)

2. “For our second analysis, we re-coded 23 characters for *Jeholotriton* and 33 characters for *Iridotriton* (Appendix A, Section 3), based on details in relevant publications (Evans et al., 2005; Wang and Rose, 2005; Carroll and Zheng, 2012), whereas both taxa were excluded in preceding study (Jia and Gao, 2019)” (Rong et al., 2021: p.720)

We are happy to talk about our interpretations of the cladistic analyses done by Rong et al. (2021) and show how their results do not impact our work: Rong et al. (2021) conducted a total of five different phylogenetic analyses, and their Figure 5A and Figure 5B show the result of their first and second analyses, respectively. By taking a very close look at Figure 5, we see that not only were *Beiyanerpeton* and *Qinglongtriton* not recovered as basal salamandroids, but neither Hynobiidae nor Cryptobranchoidea were recovered as monophyletic groups. These results from Figure 5 of their work certainly invite skepticism because not only have *Beiyanerpeton* and *Qinglongtriton* been supported as basal salamandroids by many diagnostic characters of Salamandroidea as recovered from our original cladistic analyses (Gao and Shubin, 2012; Jia and Gao, 2016; see other responses below), but also the monophyly of both the Hynobiidae and Cryptobranchoidea have been well established by studies using molecular data alone, morphological data alone, and studies combining both molecular and morphological data (summarized in our work on *Nuominerpeton*; https://peerj.com/articles/2499/).

Rong et al. (2021:p.720; or at the bottom left on the page with Figure 7) also expressed their lack of confidence in the results of the first two cladistic analyses by saying: “Because the family-level relations of living salamanders recovered in our first two analyses are inconsistent with molecular results (cf. Figure 5A, B versus Figure 6B), we performed a third analysis in which relationships of recent families were constrained by the molecular tree (Pyron and Wiens, 2011, Figure 6B)”.

As shown in Figure 6A of Rong et al., 2021, their third analysis not only recovered *Qinglongtriton* and *Beiyanerpeton* as basal members of Salamandroidea, *Regalerpeton* as stem hynobiid, but also successfully found the monophyly of Cryptobranchoidea, although hynobiids still failed to form a clade as admitted by Rong et al., 2021 (see P.720, or the paragraph on the right side of Figure 7). *Chunerpeton* was found as forming a polytomy with two other clades of crown urodeles. These results are more consistent with previous studies including Rong, 2018. Their fourth (as shown in their Figure 6C) and fifth (as shown in their Figure 7) analyses were conducted following the same strategy as in their third analysis, but most fossil salamander taxa including *Beiyanerpeton* and *Qinglongtriton* were excluded, and therefore these results are not helpful in understanding the real phylogenetic position of many fossil taxa including *Chunerpeton*.

Rong et al. (2020) further pointed out the nomenclatural fact that "herpeton" is grammatically neuter and that therefore the International Code of Zoological Nomenclature automatically corrects a number of species names from "-is" (masculine or feminine) to "-e" (neuter). The authors and dates of the names are not affected by this.

That is correct. We cited Rong et al., (2021) for this purpose in our Materials and methods section as below: “To keep the gender of species names consistent with that of genus names as per ICZN codes, we replaced the feminine/masculine species ending (“-is”) by corresponding neuter forms (“-e”) for genus names (e.g., *Nuominerpeton*) end with the neuter noun “herpeton” or “ἑρπετόν” in Greek as suggested in Rong et al. (2021).”

Species named after localities usually ends with “ensis” and does not create problems when the ending of genus names is either feminine or masculine, because “ensis” has the same form for both feminine and masculine. Considering that the Greek word “herpeton”, “ἑρπετόν”, is neuter in grammar and when it is appended to the genus name the corresponding species name should be neuter as well. The neuter form of the ending “ensis” is “ense” and now we have revised the species name ending with “ensis” by “ense” for genus names ending with “erpeton” throughout the manuscript and the supplementary file 1, including: *Beiyanerpeton*, *Chunerpeton*, *Nuominerpeton*, *Pangerpeton*, and *Regalerpeton*.

You follow common usage among paleontologists since the early 1990s in calling the crown group of salamanders Urodela and the total group Caudata. Apparently without talking to anyone, Wake (2020) has defined the name Caudata as applying to the crown group of salamanders in a way that is valid under the International Code of Phylogenetic Nomenclature (Cantino and de Queiroz, 2020). Wake (2020) explicitly left the name Urodela undefined. It might be best if you mention this situation in a few words in the manuscript. In the longer run, beyond this manuscript, it would probably be best to ask the Committee on Phylogenetic Nomenclature for an emendation of the definition of Caudata. I'm a member of the Committee and would happily coauthor a paper for this purpose with you and other experts on salamanders.

As you said, most researchers are now using Caudata to represent the total group salamanders and Urodela the crown group salamanders, and only few works used the term the other way around (e.g., Schoch, 2020). All of our four text figures have explicitly labeled total group salamanders as Caudata and crown group as Urodela, and we feel no need to stress/advertise somewhere in our text the “not popular” way of usage proposed by Wake (2020). ICPN is gaining momentum in the Era of cladistics and we appreciate your invitation to coauthor a paper to address the usage of Caudata/Urodela. Let’s stay in touch and chat over ideas on how to proceed with this project.

Lines 30, 40: I don't think Chinlestegophis and Rileymillerus should be accepted as undoubted caecilians. But as it happens, an undoubted Late Triassic caecilian was recently announced in a published conference abstract by Kligman et al. (2021), so the oldest known caecilians are Late Triassic in age either way.

Thank you for letting us known the new discovery of another Late Triassic caecilian. We kept the citation “Pardo et al., 2017a” (which is the original study proposing the stem caecilian affinities for *Chinlestegophis* and *Rileymillerus*) in the first sentence of Introduction Section, and at the same place we added “Kligman et al., 2021” and updated the reference list to strengthen our statement of the oldest known caecilian dates back to the Triassic.

35: These two papers are neither the most recent nor in any other sense the most important ones on this subject. I recommend citing Pardo et al. (2017b) and Daza et al. (2020: Figure 4D, E, S13, S14) instead.

We kept the original two citations (Fröbisch and Schoch, 2009; Maddin and Anderson, 2012) because both papers are relevant in paleoecological interpretations of dissorophoid temnospondyls, which are really our focus in this sentence as well as this manuscript. We also added “Pardo et al., 2017b” here as suggested, however we did not add “Daza et al., 2020” because there are so many uncertainties in their numerous phylogenetic hypotheses, for instance salamander species *Chelotriton*, karaurids and other urodeles were often recovered as not forming a monophyletic clade (figures 4E, 4F, S14, S15). Most importantly, Daza et al., 2020 contains no information on the paleoecological interpretations of purported ancestor groups of lissamphibians.

36: "2017b" is an error for "2017a".

Our mistake. We replaced “2017b” with “2017a”.

37: The review paper by myself and Laurin (2013) is not up to date, being written long before completion of the large phylogenetic analysis by myself and Laurin (2019), let alone its update by Daza et al. (2020: Figure 4F, S15), not to mention the analysis of ontogeny by Laurin et al. (2022). Importantly, it is clear that Lissamphibia is not derived from aquatic lepospondyls.

As you can tell in this sentence what we are really emphasizing is the paleoecological interpretations of the potential ancestral groups of lissamphibians, and you and Laurin’s work published in 2013 contains sufficient information (both information on hypotheses on Lissamphibia origin and paleoecological preferences of lepospondyls) we needed. But here we are happy to keep our work more up-to-date by citing your other works (Marjanović and Laurin, 2019; Laurin, Lapauze and Marjanović, 2022).

40-42: This is plainly not true. Chinlestegophis obviously lived in burrows, but apart from its elongate body shape it shows no adaptations to burrowing. Even the Early Jurassic Eocaecilia and, as far as its fragmentary remains allow us to tell, the Early Cretaceous Rubricacaecilia lack some of the crown group's adaptations to burrowing, e.g. they retain limbs, larger orbits and more sutures in the skull.

Thanks for letting us know your different opinion on *Chinlestegophis*’s morphological adaptations for burrowing. However, besides body elongation several other morphological specializations to support the burrowing mode of life for *Chinlestegophis* were listed in the original study (cited in our manuscript as Pardo et al., 2017a: p. E5393): “the consolidation of the skull, reduction of the orbits, and anteriorization of the jaw articulation suggests that Triassic stem group caecilians were increasingly specialized for life and feeding in confined spaces”. To remove any confusions, we revised this sentence as “…caecilian *Chinlestegophis* from the Triassic have displayed several morphological specializations as their living relatives…”

49: Of any two sister groups, each is more primitive than the other in some respects but not in others. It makes little sense to call Cryptobranchoidea "the most primitive clade of the crown group salamanders". I suggest "the sister group to all other crown salamanders".

We agree with your first sentence in the comment but our original sentence at Line 49 has nothing to do with the sister group of Cryptobranchoidea, instead the sister-group taxa within Cryptobranchoidea is what we were focusing on.

The crown group salamanders, Urodela, are traditionally classified into three suborders: Cryptobranchoidea, Sirenoidea and Salamandroidea. Our comparison was based on these three clades. Cryptobranchoidea is usually found as the sister-group taxon to Sirenoidea + Salamandroidea, but Sirenoidea is sometimes found by other studies as nested within Salamandroidea in different places. Moreover, Cryptobranchoidea has many more plesiomorphic features (e.g., two centralia in the mesopodium) than Sirenoidea and Salamandroidea. In this regard and considering the uncertain phylogenetic positions of Sirenoidea, we kept the sentence the way it is.

57, 625-626: For the life history of Aviturus, see Skutschas et al. (2018).

Thank you for reminding us of this paper. The main conclusion of Skutschas et al. (2018)’s work on the zygapophyseal skeletochronology of *Aviturus* is that this fossil taxon shares with modern cryptobranchids by having a similar growth rate. This conclusion took us a step closer to the lifestyle of *Aviturus*, however neither the ecological preference nor life history strategy (neoteny/metamorphosis) of *Aviturus* was explicitly stated. We added this citation at Line 57 and Lines 625-626 after “Vasilyan and Böhme, 2012” and updated our reference list.

58-59: It is not excluded that Regalerpeton is a stem-group salamander (Rong et al. 2020: Figure 5).

see our above responses.

71, 73: Two more good opportunities to cite Rong et al. (2020).89, 173-179, 446-450, 615: but see Rong et al. (2020) for reasons for skepticism.

For the second comment please see our responses above. Below are our responses to the first comment.

The reason we refused to cite Rong et al. (2021) at the suggested places is that we can clearly see that the real problem of Rong et al., 2021’s redescription on the so called “*Chunerpeton tianyiense*” was mainly based on “31 referred fossil skeletons” (see P.709 in the Materials and methods Section) that have a “snout-pelvic length ranging from 20 mm (IVPP V 13241A&B) to 115 mm (IVPP V 15422)” (see P. 710 in the first paragraph of section “4.1 General features”). As you can tell from their Material and methods section, most specimens are incomplete (see Rong et al., 2021: p.710). Indeed, their anatomical interpretations and line drawings are mainly based on a juvenile form (IVPP V13343; see Rong et al., 2021: figures 1 and 2) with a snout-pelvic length of about 90 mm and a total length less than 110 mm, and several other even smaller specimens (IVPP V 14226A as in Figure 3A; IVPP V 15422 as in Figure 3B) if you compare their skull length as shown in Figure 2 and Figure 3A and 3B. The specimen IVPP V13343 and other specimens display juvenile features, including but not limited to the presence of a frontoparietal fontanelle, weakly ossified and loose articulation patterns seen in the phalanges (Figure 3G, 3H), and the weakly ossified epiphysis in the hindlimb (Figure 3I, 3J). Moreover, an ossified orbitosphenoid was admitted (Rong et al., 2021:p.715, at the bottom left) to “form the bony lateral wall of braincase in most mature salamanders except proteids (Rose, 2003)” and such an ossified orbitosphenoid is “not observed…in any of our specimens”. By contrast, the holotype specimen (total length ~ 180 mm) of *Chunerpeton tianyiense* lacks the frontoparietal fontanelle, has an ossified orbitosphenoid (termed as “hypohyal” in type description) and has the humerus almost in contact with the radius/ulna. It is also important to note an ossified orbitosphenoid is absent in the 46 reported specimens of basal salamandroid *Qinglongtriton* including many fully-grown adults, reinforcing that the ossification of orbitosphenoid in early salamanders is taxonomically informative (unlike the consistent presence in modern salamanders as stated by Rose, 2003), not to mention the so many other anatomical differences in *Chunerpeton*, *Beiyanerpeton* and *Qinglongtriton* (such as presence/absence of spinal nerve foramina in the vertebrae, which are important feature to differentiate Cryptobranchoidea, Salamandroidea and Sirenoidea; see Jia and Gao, 2016).

Salamander morphologies are greatly impacted by development, and only those morphological features in adults are trustworthy for taxonomic and phylogenetic interpretations. As pointed out in our previous review work (Gao et al., 2013), neotenic taxa tend to be large in body size. It is unfortunate to see Rong et al., 2021 draw their conclusions based on juvenile specimens, while reading that “*Chunerpeton* is a large fossil salamander, with some individuals reaching body lengths of up to 50 cm (~18 cm in the holotype)” in another paper (Sullivan et al., 2014: p.250) co-authored by the senior author in Rong et al., 2021. Based on so many anatomical differences from the holotype of *Chunerpeton*, it is quite likely that the batch of specimens studied by Rong et al., 2021 represent a new species that was misclassified as *Chunerpeton tianyiense*.

Anyway, we are reluctant to distract our present study by including so many uncertainties introduced by Rong et al., 2021’s work to test their conclusions. But we feel obligated to make clarifications and will hopefully address these problems with our own specimens of *Chunerpeton tianyiense* in the near future, but this is not within the scope of our current study.

References not listed in the manuscript:

Sullivan, C., Y. Wang, D.W.E. Hone, Y. Wang, X. Xu, and F. Zhang. 2014. The vertebrates of the Jurassic Daohugou Biota of northeastern China. Journal of Vertebrate Paleontology 34:243–280.

110, 198, 268, 271, 276, 277, 279, 280, 282, 284, 286, 289, 290, 334, 337, 428: Insert "last" before "common"!

Good point, done.

194: Replace "albert" by "albeit" (or "although" or just "though").

Our typo. Thank you!

202, 602, 635: Uppercase for Procrustes (the name of a mythological person).

“p” is now capitalized for “Procrustes” throughout the text.

293-294: All adult anurans and caecilians are metamorphosed, but some of both are fully aquatic (e.g. Pipidae, Typhlonectidae). I would therefore rearrange the sentence to: "Among modern amphibians, the adults of anurans and caecilians are metamorphosed, and most of them are terrestrial."

Thanks for the suggestion. We refused to add “the adults of anurans and caecilians are metamorphosed” considering that the life history strategy of many caecilians is direct development, which is different from metamorphosis because there is no two-phased development after hatching for direct developers. We kept the original sentence and added “mostly” between “their postmetamorphosed adults are” and “terrestrial” to cover the exceptions of the few aquatic taxa.

299: …or whatever will remain of Lepospondyli.308: I would write "metamorphic taxa" or "metamorphosed individuals".

We chose the latter suggestion.

339-340: *Aviturus*, which dates from the very end of the Paleocene, is not the earliest known pancryptobranchan. The earliest entirely undoubted one is "*Cryptobranchus*" *saskatchewanensis*, which is a few million years older (late middle Paleocene). There is evidence that the much older (mid-Cretaceous) Eoscapherpeton is a stem-pancryptobranchan; see Marjanović (2021: supplementary material pp. 13-16) for references and a brief review.

Thanks for the correction and reminding of your summary on *Eoscapherpeton* and other fossil cryptobranchids. We now have the sentence “The earliest known fossil pancryptobranchan *Aviturus*” replaced as “The Paleocene pancryptobranchan *Aviturus*”.

480: The plural of pars palatina is partes palatinae – as in most languages of Europe (but unlike English), number and gender (and case) are marked on adjectives as well as nouns in Latin.

Thank you so much for correcting our usage on the plural form of “pars palatina” from “partes palatina” to “partes palatinae” and please find more of our responses to this issue in our replies to Reviewer #1.

484: Replace "pas" by "pars".

Thanks for pointing out the typo. We replaced “pas” by “pars”.

791: Replace "batrachian" by "batrachians".

done.

857-859: Unlike English, German does not have separate capitalization rules for headlines – but it always gives a capital Letter to every Noun, while almost nothing else is ever capitalized, not even Adjectives derived from proper Names. Therefore, please correct the Title to: "Generelle Morphologie der Organismen: Allgemeine Grundzüge der organischen Formen-Wissenschaft, mechanisch begründet durch die von Charles Darwin reformirte Descendenz-Theorie".

Revised. Good to know.

Finally, the supplementary file claims to be a PDF file ("This PDF file contains supplementary file 1A to 1K."), but it is a DOCX file. In the interest of wider accessibility, I recommend you convert it to PDF before resubmitting it.

Done.

I hope these comments are helpful. I apologize again for the delay, and I'm looking forward to the next version of your manuscript!References not cited in the manuscript:Cantino PD, de Queiroz K. 2020. International Code of Phylogenetic Nomenclature (PhyloCode). Version 6. CRC/Taylor & Francis/Informa, Boca Raton/London/New York. ISBN: 978-1-138-33282-9 (paperback), 978-1-138-33286-7 (hardback), 9780429446320 (e-book). DOI: https://doi.org/10.1201/9780429446320 Openly accessible at http://phylonames.org/code/Daza JD, Stanley EL, Bolet A, Bauer AM, Arias JS, Čerňanský A, Bevitt JJ, Wagner P, Evans SE. 2020. Enigmatic amphibians in mid-Cretaceous amber were chameleon-like ballistic feeders. Science 370:687-691. DOI: https://doi.org/10.1126/science.abb6005Kligman B, Stocker M, March A, Nesbitt S, Parker W. 2021. New Late Triassic stem-caecilian from southwestern North America strengthens evidence for lissamphibian monophyly, and illuminates the anatomical, functional and geographic origins of living caecilians [abstract]. Society of Vertebrate Paleontology (ed.): Virtual meeting conference program, 81st annual meeting, p. 160. The entire abstract volume can be downloaded here: https://vertpaleo.org/svp_2021_virtualbook_final/Laurin M, Lapauze O, Marjanović D. 2022. What do ossification sequences tell us about the origin of extant amphibians? Peer Community Journal 2:e12. DOI: https://doi.org/10.24072/pcjournal.89Marjanović D. 2021. The making of calibration sausage exemplified by recalibrating the transcriptomic timetree of jawed vertebrates. Frontiers in Genetics 12:521693. DOI: 10.3389/fgene.2021.521693Marjanović D, Laurin M. 2019. Phylogeny of Paleozoic limbed vertebrates reassessed through revision and expansion of the largest published relevant data matrix. PeerJ 6:e5565. DOI: https://doi.org/10.7717/peerj.5565Rong Y-F, Vasilyan D, Dong L-P, Wang Y. 2020 (printed 2021). Revision of Chunerpeton tianyiense (Lissamphibia, Caudata): Is it a cryptobranchid salamander? Palaeoworld 30:708-723. DOI: https://doi.org/10.1016/j.palwor.2020.12.001Skutschas PP, Kolchanov VV, Bulanov VV, Sennikov AG, Boitsova EA, Gulbev VK, Syromyatnikova EV. 2018 (printed 2020). Reconstruction of the life history traits in the giant salamander Aviturus exsecratus (Caudata, Cryptobranchidae) from the Paleocene of Mongolia using zygapophyseal skeletochronology. Historical Biology 32:645-648. DOI: https://doi.org/10.1080/08912963.2018.1523157Wake DB. 2020. Caudata J. A. Scopoli 1777 [D. Wake], converted clade name. [Brackets in the original.] de Queiroz K, Cantino PD, Bauthier JA (eds): Phylonyms. A Companion to the PhyloCode (CRC/Taylor & Francis/Informa, Boca Raton/London/New York), pp. 785-787. ISBN: 978-1-138-33293-5 (hardback), 9780429446276 (e-book). DOI of the entire book: https://doi.org/10.1201/9780429446276

[Editors' note: further revisions were suggested prior to acceptance, as described below.]

The manuscript has been improved but there are some remaining issues that need to be addressed, as outlined below:Reviewer #3:The issues with image quality in the merged PDF have disappeared. Please correct the species names in figures 1, 3, 19, 23-26, 30-32, 36 and 38 if I've counted correctly, and replace "semiaqaic" with "semiaquatic" in Figure 3.

Correct species names are now labeled in Figures 1, 3; and the many supplementary figures: Figure 1—figure supplements 15, 19-22, 24, 26; Figure 2—figure supplements 1, 5; and Figure 3—figure supplement 2. Our typo was also corrected in Figure 3.

Sirenidae and the first crown-group salamander.Your anatomical reasons for the omission of sirenids are convincing, and I had indeed confused sirenids and amphiumids in my statement about partial metamorphosis – thank you for pointing this out! I am, however, quite surprised that you present the phylogenetic position of Sirenidae as a mystery. Sirenidae and Salamandroidea have been found as sister groups in every large-scale study of molecular data, no matter which data or which method exactly: 7,189 transcripts of nuclear genes and Bayesian inference (Irisarri et al. 2017; 99% posterior probability), 5 mitochondrial and 10 nuclear genes and maximum likelihood (Vijayakumar et al. 2019: supplementary file Amphibia_New_India_SHL_Dryad.tre; 52% bootstrap support – note this is the latest and largest version of R. A. Pyron's series of matrices, and all versions, including Jetz and Pyron [2018] which you cite, found the same topology), or 120 nuclear protein-coding genes, separately and together, and maximum likelihood (Himes et al. 2020: 100% bootstrap and ASTRAL posterior values). Given such extraordinary agreement, the fact that the morphological evidence is less clear can be blamed on the morphological evidence, or on our lack of knowledge of it (I greatly appreciate your recent work in describing the osteology of extant salamanders!), but it would be very difficult to try to argue the molecular data away. Therefore, I think you should mention in the text that you cannot include Sirenidae in your dataset for the reasons you presented in your response and that this is a potential weak point in inferences about the last common ancestor of all crown-group salamanders.

We thank you for your summary on the recent molecular cladistic analyses and their conclusions on the phylogenetic position of sirenids. Based on our last responses, we now have added a few sentences in the first section of Materials and methods titled “Experimental design, specimens, palate” to explain why sirenids were not included in our dataset.

You point out that the analyses of Rong et al. (2020, printed 2021) failed to find Hynobiidae or Cryptobranchoidea. Unfortunately, this is not incompatible with their matrix being an improvement over its sources. It is not uncommon (e.g. Marjanović and Laurin, 2019, and references therein) for improvements to matrices for phylogenetic analysis to decrease resolution, and it is not uncommon either for matrices to support the right things for the wrong reasons. Matrices that support clades for the wrong reasons are unreliable in the sense that they cannot be trusted to place added taxa accurately. As I stated, the version by Rong et al. (2020) cannot be relied upon as strong evidence for the phylogenetic positions of Chunerpeton, Beiyanerpeton or any other taxon – but that does not make the earlier versions of that matrix any better. They contain, after all, such phenomena as redundant and even duplicate characters: for example, the haploid and the diploid number of chromosomes are both included as separate characters in the previous versions. Duplicating a character inevitably distorts, if not the topology, then at the very least the support values for that topology (references in Marjanović and Laurin, 2019: 15-16). Rong et al. (2020) made a certain effort to reduce this problem.They also ordered certain characters, unlike their sources. Ordering is widely believed to be a philosophical question, but there is strong evidence from simulations as well as from empirical studies that potentially clinal or meristic characters must be ordered to avoid inaccurate results as well as both false negatives and false positives in resolution (references and brief discussion in Marjanović and Laurin, 2019: 16).

The matrix of Rong et al., 2021 is an improvement of the source dataset (Jia and Gao, 2019 which in turn is based largely on Gao and Shubin, 2012) in terms of recoding certain characters for *Jeholotriton*, *Iridotriton* and *Chunerpeton* based on their own reinterpretations and inclusion of molecular trees as backbones. We checked this paper carefully and, unfortunately, we didn’t find they “ordered certain characters” because they stated it clearly in the first paragraph in Section 6.3---“We designated the stem salamander *Karaurus* as outgroup and set all characters as unordered and equally weighted” (page 719). But experimenting analytic strategies like ordering certain characters as you mentioned above should definitely be encouraged, especially when developmental trajectories of certain characters are well understood. Therefore, more studies on relevant extant taxa and growth series as we are working on will be potentially helpful to increase sampling in taxa and characters for cladistic analyses.

Our original arguments in previous responses are that their results from the first two cladistic analyses (where monophyletic status for Hynobiidae and Cryptobranchoidea was not recovered) of Rong et al. (2021) were even doubted by the authors themselves, and their several inferences including the one you mentioned in your previous comments (*Beiyanerpeton*, *Qinglongtriton*, possibly all other Mesozoic Chinese salamanders outside the salamander crown group) are paradoxically based on their first two cladistic analyses. However, these inferences can not be drawn from their third cladistic analysis, which includes a molecular tree as backbone and is also favored by the authors.

Phylogenetics with morphological dataParsimony analysis of morphological data should not be treated as a black box.Daza et al. (2020) did all their phylogenetic work on previously published matrices; they added or updated the scores of Albanerpetidae, but did not make any other changes to those matrices. Therefore, practically all the problems with their results are the fault of those matrices. The clearly wrong placement of Chelotriton in their Figure 4F and S15 stem from the fact that I had added Chelotriton to the source matrix in question, that of Marjanović and Laurin (2019), which is simply not equipped to handle crown-group salamanders. (That matrix is an attempt to improve the one by Ruta and Coates, 2007. In order to limit the amount of time and effort, we did not add any characters to that matrix. The matrix of Ruta and Coates, 2007, contained only two salamanders – Karaurus and Valdotriton – and therefore lacked any characters specific to salamander phylogeny.) I did this in order to test a point on ontogeny and phylogeny: would the extreme metamorphosis of Chelotriton, which makes it look more like a temnospondyl than lissamphibians usually do, pull some or all lissamphibians into Temnospondyli? That did not happen, but Chelotriton was pulled out of the salamander clade. The latter result is obviously wrong, and obviously due to the lack of salamander-specific characters in the matrix; there is no reason to think that it indicates a more general problem. The former result, on the other hand, is actually strengthened that way.In another analysis, Daza et al. (2020: Figure 4E, S14) found Karauridae outside Batrachia. I haven't looked into its precise causes, but it is simply shared with the source matrix of that analysis, the matrix of Pardo et al. (2017a).Both of these analyses are, however, relevant to the ecology of the first lissamphibian by finding Albanerpetidae as the sister group of Lissamphibia, Albanerpetidae being terrestrial and fully metamorphic or of course direct-developing (early juveniles of any sort are not known). This is a data point for phylogenetic bracketing.

Thank you for sharing with us the backstories of the cladistic analyses done by Daza et al., 2020.

In the context of the specimens referred to Chunerpeton by Rong et al. (2020), you write: "only those morphological features in adults are trustworthy for taxonomic and phylogenetic interpretations." In phylogenetics, however, morphological features are not automatically comparable just because they are found in sexually mature adults, as you seem to have assumed in your published phylogenetic work: for much of the skeleton, for instance, neotenic adults are much more easily comparable to larvae than to adults of metamorphic taxa, so that the presence or absence of many character states of metamorphic adults must be scored as unknown/inapplicable in neotenic taxa. This has a strong effect on the topologies found by phylogenetic analyses (Wiens et al., 2005; Marjanović and Laurin, 2019: 21-22, and references therein).

What we argued are that development in salamanders have enormous impacts on their morphological features, and states of characters in larval/juvenile/subadult specimens will eventually be replaced or at least modified by states of characters observed in sexually mature specimens, and therefore the final, stable state of a character is only found in sexually-mature specimens regardless of whether the species is neotenic, metamorphosed or direct developing. Even in neotenic salamanders, in which morphological changes during development are fewer than that in metamorphosed or direct developing taxa, certain characters such as proportion, shape, ossification extent will change developmentally and we need to maximally alleviate effects on phylogeny/taxonomy from ontogeny by scoring characters based on adult specimens.

Comparing character states displayed by adult specimens from species in different lifestyles (e.g., neoteny, metamorphosis), as what you emphasized above, is indeed a different story. We agree that certain morphological features are retained from larvae to adults in neotenic taxa and otherwise are resorbed during metamorphosis and thus cause incomparable phenomena for neotenic and metamorphic/direct developing taxa. But these phenomena should not be deemed as reasons to not score characters from adult specimens but instead should be creatively dealt with analytical strategies such as coding neotenic characters as “?” or character ordering or weighting.

In this case, the so-called “referred specimens” to *Chunerpeton tianyiense* are all juveniles based on several evidences we listed in previous responses (e.g., incomplete ossification of epiphyses of long bones). Without adult specimens, the taxonomic affiliation of these juveniles is truly unknown. One possibility as we mentioned in our previous response is that these juveniles may represent a new species of *Chunerpeton* because they have displayed differences with the holotype specimen of *C*. *tianyiense* (e.g., orbitosphenoid present & absent). Another possibility is that theoretically theses juvenile specimens may simply represent not-yet-metamorphosed-individuals of an unknown, metamorphic species, because neotenic taxa in fossil record can only be distinguished by the presence of larval features in adult specimens.

Nomenclature and taxonomyI greatly appreciate your comments on the validity of the referral of the new specimens by Rong et al. (2020) to Chunerpeton, and I'm looking forward to your further publications on this matter!Likewise, I appreciate your comments on the definitions of the names Caudata and Urodela and will try to send you a draft manuscript as soon as possible. For the purposes of the present manuscript, I withdraw all my objections.

Thank you for your invitation and we look forward to working with you on this interesting manuscript! We have to say, as you may have already noticed, that Evans and Milner (1996) have provided a brief “Taxonomic note” in their paper (p. 629) describing *Valdotriton* explaining the usage of “Caudata” and “Urodela” before 1988 and after 1988. We will be prepared for your manuscript and try to dig up the old literatures for the usage of these two terms.

Evans SS, Milner AR. 1996. A metamorphosed salamander from the Early Cretaceous of Las Hoyas, Spain. *Philosophical Transactions: Biological Sciences* 351:627-646. URL: http://www.jstor.org/stable/56320

Adaptations of ChinlestegophisThank you for reminding me of the features listed as adaptations to burrowing by Pardo et al. (2017a). I agree that the fairly small orbits may count as such, though they could also hint at life at the bottom of muddy bodies of water – a strictly aquatic lifestyle, apart from sheltering in burrows with 100% humidity, is assured by the lateral-line grooves. The position of the jaw articulation, however, is shared with all other brachiopods, where it is an adaptation to a particular style of suction feeding; it is in fact more extreme in Batrachosuchus, depicted in Pardo et al. (2017a: Figure 3). By "consolidation of the skull", Pardo et al. (2017a: supp. inf. part F) mean the frankly irrelevant fusion of lacrimal and maxilla – besides, it is not in fact clear whether a lacrimal is even present -, the supposed fusion of the pterygoid and the quadrate for which there is very little evidence presented in the paper, and fusion of the exoccipitals and the basioccipital which is universal in dissorophoid temnospondyls, none of which were burrowing or had any sort of digging lifestyle – and note that it is not clear if a basioccipital was present in the first place; it remained cartilaginous and very small in other brachiopods. In a borrower, one would expect, as in extant caecilians, a well-ossified braincase that could function as a strut to enable the skull to resist rostrocaudal compression, dorsoventral bending, or twisting; yet, most of the braincase is not ossified at all in Chinlestegophis (Pardo et al. 2017: supp. inf. part B). This is particularly striking in comparison to the burrowing "lepospondyls" described by Pardo and various coauthors in the two years prior.Interestingly, the stem-caecilian presented by Kligman et al. (2021) seems not to be adapted to burrowing at all, but it is clearly much closer to the crown group than Chinlestegophis.

Thank you for sharing with us your thoughts on the burrowing adaptation in early caecilians.

DetailsLines 38-41: I had not quite appreciated that here you cite references both for the lifestyles and the phylogenetic positions of the listed taxa, and only suggested references for the phylogenetic positions of various "lepospondyls". For the terrestrial lifestyle of at least one amphibamid, I recommend Laurin et al. (2004) and references therein. For the various lifestyles of "lepospondyls", I recommend Jansen & Marjanović (2021) and references therein. While some stereospondyls have occasionally been considered semiaquatic for vaguely articulated reasons, almost all were certainly fully aquatic as shown by the lateral-line grooves on their skulls and further supported by their very slow peri- and endochondral ossification; I recommend the brief but clear statement in Schoch & Milner (2014: 123) and references therein. Notably, Chinlestegophis has lateral-line grooves (Pardo et al., 2017a), showing that it was not semiaquatic; "reduced" grooves as identified by Pardo et al. (2017a) mean that most of the lateral-line organ was situated in the skin and did not contact the bone, not that the organ was "reduced" – something that does not occur, because the lateral-line organ dries up and dies from serious exposure to air as inevitably caused by a semiaquatic lifestyle. – While Jansen & Marjanović (2021) is published on a preprint server with the usual disclaimer, it is in fact an accepted manuscript published with permission from the journal (Comptes Rendus Palevol) that accepted it after peer review. The journal is currently changing publishers, a process that started before the pandemic and is still not complete; we've been waiting for the page proofs since March 2021.

Thank you for providing us more literatures on the lifestyles of early tetrapods. We now have added “Laurin et al., 2004”, “Jansen and Marjanović, 2021” and “Schoch and Milner, 2014”. We also replaced “semiaquatic” by “semiaquatic/aquatic” for stereospondylian at Line 40.

80-81: Other than its probable membership in Karauridae, is there evidence on the lifestyle of Marmorerpeton?

As we originally stated here *Marmorerpeton* was argued to be neotenic by Evans et al., 1988 and many later studies especially those done by Dr. Pavel Skutschas and his colleagues.

85, 106: If you use Caudata as the name for the total group, Triassurus is a stem caudate, not a stem urodele; nothing is a stem urodele, because if something is on the stem from which Urodela comes, it is outside Urodela by definition. At least for the first few decades of this terminology, a clade consisted of a crown group and a stem-group; to be a stem urodele, something has to be a urodele.

We do not agree with the above argument. We tend to believe Caudata (total group) and Urodela (crown group) are two node-based clades. Many taxa including those we included in the Introduction, such as *Marmorerpeton*, *Karaurus*, *Urupia* are stem urodeles because they lack the spinal nerve foramen on atlas that is otherwise present in Urodela. *Triassurus* is a salamander and it does not belong to the crown group salamander, Urodela, but is located within the total group salamander, Caudata. So we can call it basal caudate or stem urodele to ensure people understanding its rough phylogenetic position. If we designate *Triassurus* to be stem caudate as suggested here, it then will not a salamander but instead a “salamander-morph”. Unfortunately, in the PNAS paper of Schoch et al. (2020) where *Triassurus* was restudied based on a second larval specimen, Caudata and Urodela were designated as crown and total group salamanders, respectively, and again unfortunately, *Triassurus* was called to be a “stem caudate”. Moreover, because of the larval stages of both the holotype and the second specimen, *Triassurus* remains enigmatic in many aspects (e.g., life history strategy), and states of many characters in fully-grown specimens of *Triassurus* remain unclear, including the presence/absence of spinal nerve foramina on atlas.

106: Likewise, if you use Urodela as the name for the crown group, "crown urodeles" is redundant; "urodeles" would be enough.

The original sentence is “based …on the palate of all living and …fossil genera of cryptobranchoids, stem and other basal crown urodeles ….”. The reason we kept the word “crown” is we want to make it clear to the readers that we investigated both stem urodeles and basal crown urodeles.

623-624: I would rather write: "As implied by Jia et al. (2021a), we apply the name Hynobiidae to the crown group of Panhynobia".

We replaced “The family Hynobiidae is designated here as crown group Panhynobia” with “Following Jia et al. (2021a), we apply the name Hynobiidae to the crown group of Panhynobia”.

912: "lepospondyl", not "lepospondyls".

The second “s” in “lepospondyls” is now deleted.

928: "Kuro-o", a Japanese name with three syllables; "oo" could be misunderstood as a long vowel – Japanese distinguishes long from short vowels. Sometimes apostrophes are used to disambiguate in transcriptions, sometimes hyphens are used instead.

We replaced “Kuro-O” by “Kuro-o”.

976: "Syromyatnikova".

Done.

977: italics for the genus & species name (present in the original, I've checked)

Done.

Suggestions on style and languageLines 15-18: "but the small number of reliable ecological indicators established so far hinders investigations into the paleobiology of early salamanders. Here we statistically demonstrate, by using time-calibrated phylogenetic trees and geometric morphometric analysis on 71 specimens in 36 species, that both the shape"…

There are indeed small number of reliable ecological indicators in salamanders, but these characters are from soft tissues such as external gill, caudal fin, digit web in hand and foot that are rarely preserved in fossil record particularly in metamorphosed taxa. No osteological characters that are potentially informative in ecological preferences has been investigated or tested with fossil taxa except the palate studied here. Therefore, we retained the original sentence “but the yet established reliable ecological indicators from bony skeletons hinder investigations into the paleobiology of early salamanders.”

21: I'm not sure what you mean by "strictly". Perhaps "analyzed in detail" would be clearer?

“strictly analyzed” is now replaced with “analyzed in detail”.

25-27: That would mean the disparities, not the salamanders, have achieved the ecological preferences. Would the following be an improvement? "Salamanders began to diversify ecologically before the Middle Jurassic and achieved all their present modes of life in the Early Cretaceous."

We replaced the original sentence into “Salamanders are diversified ecologically before the Middle Jurassic and achieved all their present ecological preferences in the Early Cretaceous”.